# Effect of Irrigation Level and Irrigation Frequency on the Growth of Mini Chinese Cabbage and Residual Soil Nitrate Nitrogen

**Youzhen Xiang [1], Haiyang Zou [1], Fucang Zhang [1,\*], Shengcai Qiang [1], You Wu [1], Shicheng Yan [1], Haidong Wang [1], Lifeng Wu [1], Junliang Fan [1] and Xiukang Wang [1,2,\*]**

1. Key Laboratory of Agricultural Soil and Water Engineering in Arid and Semiarid Areas of Ministry of Education, Northwest A&F University, Yangling 712100, China; youzhenxiang@nwsuaf.edu.cn (Y.X.); zouhaiyang@nwsuaf.edu.cn (H.Z.); qiangsc@nwsuaf.edu.cn (S.Q.); wuyou@nwsuaf.edu.cn (Y.W.); yanshicheng@nwafu.edu.cn (S.Y.); wanghd@nwsuaf.edu.cn (H.W.); wulifeng@nit.edu.cn (L.W.); nwwfjl@nwsuaf.edu.cn (J.F.)
2. College of Life Sciences, Yan'an University, Yan'an 716000, China
* Correspondence: zhangfc@nwsuaf.edu.cn (F.Z.); wangxiukang@yau.edu.cn (X.W.); Tel.: +86-029-87091151 (F.Z.); +86-0911-233-2030 (X.W.)

**Abstract:** Nitrogen (N) fertilizer is known to improve the quality and biomass of vegetables, but it is unclear how to manage the large amount of $NO_3$-N that accumulates in the soil after vegetable harvest. In this study, we examined the influence of irrigation level and irrigation frequency on the growth and soil residual $NO_3$-N of the catch crop mini Chinese cabbage (*Brassica pekinensis*) in a greenhouse. Using conventional border irrigation with adequate water supply as a control (CK), three irrigation levels ($W_H$: 160% crop evapotranspiration (ETc), $W_M$: 120% ETc and $W_L$: 80% ETc) and three irrigation frequencies (intervals of $F_2$: 2 days, $F_4$: 4 days, and $F_8$: 8 days) were assessed in 2014, 2015 and 2016 in northwest China. The results showed that the weight of the leaves and leaf stalks was the primary determinant of yield, and that these are the primary N-containing vegetative organs of the plants. At the same irrigation level, the total N content of the plants increased in the order $F_8 < F_2 < F_4$. The trend in the total N content in the mini Chinese cabbage plants among different treatments was synchronized with the yield. The highest total N content in the plants was observed in the $W_M F_4$ treatment during all three years. The three-year averages of mini Chinese cabbage aboveground biomass, yield and water use efficiency (WUE) in the $W_M F_4$ treatment were 60%, 64.5% and 119.2% higher respectively than in the CK treatment. The residual $NO_3$-N content in the soil in the $W_M F_4$ treatment was only 1.3% higher than that in the CK treatment. The total N uptake in the $W_M F_4$ treatment was 79.2% higher than that in the CK treatment, and the N loss in the $W_M F_4$ treatment was 46.3% lower than that in the CK treatment. Under these experimental conditions, the $W_M F_4$ treatment can be recommended as an appropriate irrigation regime for mini Chinese cabbage under fallow greenhouse management in northwest China.

**Keywords:** mini Chinese cabbage; economic benefits; nitrate nitrogen; water use efficiency

## 1. Introduction

High water and fertilizer inputs are usually considered to be a way to obtain high yields in the production of greenhouses vegetables [1]. Nitrogen (N) fertilizer is one of the main elements affecting crop growth and is heavily applied in crop production due to its significant improvement of crop yield [2]. For example, the amount of N applied to cucumber during a single season was as high as 1958 kg N ha$^{-1}$ in the Shouguang District, Shandong [3]. The amount of N fertilizer applied each

year is almost three times the amount absorbed by vegetables [4]. In recent years, vegetables have been the main crops grown in greenhouses in China because they provide consumers with many health benefits and have high economic value [5]. Particularly in the Guanzhong region, agricultural facilities have developed rapidly in the past two decades. The planting area accounts for more than 60% of the province. However, it is common for farmers to apply excessive fertilization and poorly planned irrigation in the management process, and the soil environmental problems caused by these methods are substantial [6,7]. In the process of the growth and development of crops, the absorption and utilization of N nutrients are limited. Excessive N application can inhibit the quality and yield of fruits and vegetables [8,9], leading to soil degradation [10], including simultaneous nitrification and denitrification by ammonia volatilization and $N_2O$ and $N_2$ emissions [11] and increased nitrate content in the groundwater [12], which seriously threatens the sustainable development of the vegetable industry in facilities [13].

Many researchers have shown that the total accumulation of $NO_3$-N in the 0–2.0 m soil layer of greenhouse soil was significantly higher than that in the adjacent grain fields [14–16]. The relative content of $NO_3$-N in the 1–2.0 m soil layer of some test sites is more than 40% [17], and the content of $NO_3$-N in the groundwater of some areas is close to the limit of drinking water sanitation standards (World Health Organization, 8 mg $L^{-1}$). The content of $NO_3$-N in the groundwater has approached the permissible limit of 6.4 mg $L^{-1}$ in the 5.25–5.5 m soil layer in the Guanzhong region [18]. The roots of catch crops such as ryegrass and fodder radish can effectively absorb the remaining $NO_3$-N in the soil [19]. Moreover, after harvest, the soil $NO_3$-N accumulation of 0–1.0 m deep soil planted with leeks and red beets was lower than that of soil without catch crops [19,20]. After the maize harvest, the loss due to nitrate leaching of the test field could be reduced by 80% by planting rye compared to that in a fallow field [21]. The accumulation of nitrate in the root layer of the soil profile could be reduced by applying catch crops in vegetable rotation systems [22,23]. Studies have indicated that planting catch crops is an effective measure to reduce the residual $NO_3$-N in the soil. In particular, soil residual $NO_3$-N was absorbed and utilized differently by different crops, and the N absorption amount of mini Chinese cabbage was up to 70 kg N $ha^{-1}$ [24].

The mini Chinese cabbage is nitrophile that is resistant to low temperatures and suitable for growing in the winter in northwest China [25]. The Guanzhong area produces relatively few vegetables in the winter, and the mini Chinese cabbage naturally has high economic value in this region. As vegetables are healthful foods, the demand for them has increased significantly in recent years [26]. Thus, local farmers can increase their income by planting mini Chinese cabbage as a catch crop. In conclusion, studying catch mini Chinese cabbage and the ways in which its nutrient use lowers residual N in field soil can improve greenhouse soil texture, help develop a scientific system of crop rotation, and increase the economic income of farmers.

Numerous studies have focused on the N absorption and soil $NO_3$-N residue of catch crops, while few reports examine the responses of these crops to the soil residual N in terms of growth, yield, quality and water utilization under different irrigation regimes. Water is the main factor influencing the yield and quality of vegetables [27,28]. Scientific irrigation methods and regimes can effectively promote crop growth and nutrient absorption [29] and control pests and diseases [30]. Drip irrigation can maintain the soil in a good condition suitable for crop growth and is conducive to high crop yield and water use efficiency (WUE) [31–33]. Studies have shown that under some irrigation conditions, the frequency of irrigation can significantly affect the growth and yield of tomato [34], pepper [35], and garlic [36]. However, the effect of irrigation level and irrigation frequency on the soil $NO_3$-N residues and growth of mini Chinese cabbage in greenhouses has never been reported.

The present study was designed to explore (1) whether the effects of irrigation levels on the growth, yield and quality of mini Chinese cabbage vary with irrigation frequency, and (2) whether the effects of irrigation levels on the WUE and soil residual $NO_3$-N migration regularity of mini Chinese cabbage vary with irrigation frequency. Answering these questions is essential for making recommendations of suitable irrigation regimes for the use of mini Chinese cabbage in fallow greenhouse management

in northwest China. In particular, this information should facilitate the establishment of scientific greenhouse catch crop irrigation systems, help to maintain the sustainable use of soil, and provide a theoretical basis for the sustainable development of agriculture.

## 2. Materials and Methods

### 2.1. Site Description

The experiments were conducted during the mini Chinese cabbage growing seasons in 2014, 2015 and 2016 at the Test Station of the Key Laboratory of the Ministry of Education for Agricultural Water and Soil Engineering in Arid Areas (108°04′E, 34°20′N) in the Guanzhong Plain in northwest China. The region is classified as warm temperate due to a local semihumid climate. The annual average temperature is 11.0 °C with an annual average evaporation of 1500 mm throughout the mini Chinese cabbage life cycle. The test greenhouse measures 76 m in length, 7.5 m in width, and 2.8 m in height. The cultivated soil layer (0–80 cm) in the experimental area is heavy soil (46% sand, 43% silt, and 11% clay). The major soil physicochemical characteristics of the experimental site measured before the experiment are shown in Table 1 (n = 5).

**Table 1.** Major soil physicochemical characteristics of the experimental site measured before the experiment.

| Years | Soil Depth (cm) | BD (g cm$^{-3}$) | OM (g kg$^{-1}$) | pH | TN (g kg$^{-1}$) | TP (g kg$^{-1}$) | TK (g kg$^{-1}$) | AP (mg kg$^{-1}$) | AK (mg kg$^{-1}$) |
|---|---|---|---|---|---|---|---|---|---|
| | 0–20 | 1.42 | 14.33 | 8.2 | 0.62 | 0.59 | 12.8 | 34.2 | 101.2 |
| 2014 | 20–40 | 1.38 | 15.44 | 8.4 | 0.76 | 0.52 | 16.8 | 21.6 | 97.6 |
| | 40–60 | 1.49 | 11.66 | 8.3 | 0.87 | 0.49 | 16.7 | 26.7 | 107.1 |
| | 60–80 | 1.43 | 13.59 | 8.3 | 0.71 | 0.53 | 15.5 | 13.5 | 91.5 |
| | 0–20 | 1.40 | 16.63 | 8.0 | 0.71 | 0.68 | 14.8 | 18.7 | 114.6 |
| 2015 | 20–40 | 1.47 | 15.88 | 8.3 | 0.65 | 0.53 | 15.6 | 15.8 | 103.3 |
| | 40–60 | 1.44 | 14.52 | 8.3 | 0.86 | 0.52 | 17.1 | 14.6 | 88.6 |
| | 60–80 | 1.37 | 13.49 | 8.1 | 0.79 | 0.42 | 16.8 | 12.6 | 94.5 |
| | 0–20 | 1.38 | 14.54 | 8.2 | 0.87 | 0.49 | 11.6 | 22.1 | 109.2 |
| 2016 | 20–40 | 1.44 | 15.87 | 8.2 | 0.79 | 0.56 | 14.7 | 16.7 | 115.8 |
| | 40–60 | 1.49 | 13.15 | 8.0 | 0.66 | 0.54 | 16.2 | 17.4 | 110 |
| | 60–80 | 1.40 | 12.98 | 8.1 | 0.75 | 0.46 | 14.4 | 10.9 | 94.4 |

BD: bulk density; OM: organic matter; TN: total nitrogen; TP: total phosphorus; TK: total potassium; AP: available phosphorus; AK: available potassium.

A small weather station (HOBO event logger, Onset Computer Corporation, USA) was set up inside the greenhouse. The temperature, atmospheric pressure, relative humidity, photosynthetically active radiation (PAR), and meteorological factors, such as solar radiation, were recorded at 10-min intervals. The data obtained confirmed that these parameters were comparable for all plots.

### 2.2. Experimental Treatments and Design

Seeds of mini Chinese cabbage (*Brassica pekinensis* cv. "Lvguan F1") were obtained from Shaanxi Xianyang Four Seasons Seedling Co., Ltd., Shaanxi, China, and sown in each field plot on the same day. The mini Chinese cabbages were grown with three irrigation levels (W$_H$: 160% crop evapotranspiration (ETc), W$_M$: 120% ETc and W$_L$: 80% ETc), which were applied at three different irrigation frequencies (F$_2$: 2-day, F$_4$: 4-day and F$_8$: 8-day intervals), and a control was designed (conventional border irrigation with adequate water supply: the moisture content of the 60 cm soil layer remained at 80 ± 5% of the field moisture capacity, CK), resulting in a total of 10 treatments (CK, W$_H$F$_2$, W$_H$F$_4$, W$_H$F$_8$, W$_M$F$_2$, W$_M$F$_4$, W$_M$F$_8$, W$_L$F$_2$, W$_L$F$_4$ and W$_L$F$_8$). Three replicates of each treatment were performed in a randomized complete block factorial design. Each field plot was 1.5 m wide and 6.5 m long with a total area of 9.75 m$^2$. The typical local row planting pattern was adopted, with an equal spacing of 6.0 cm in each row. A total of 105 healthy seedlings were obtained, and the irrigation regimes were

started at the same time. To prevent the infiltration of water from the neighboring plots, the plots were separated by embedded plastic sheets placed 1.0 m deep into the soil. The mini Chinese cabbage seeds were sown on September 20 of each year and uprooted on February 3 in 2014, February 9 in 2015, and February 5 in 2016. The crop preceding this experiment was sweet pepper in 2014 and 2015 and tomato in 2016. The same fertilizer level (N400–$P_2O_5$225–$K_2O$150 kg ha$^{-1}$), chosen on the basis of the amount of fertilizer commonly used locally, was applied to the sweet pepper and tomato. No fertilizer was applied throughout the growth period of mini Chinese cabbage. The irrigation treatments were initiated using the surface drip irrigation system during transplanting, and 15 mm of water was provided. Throughout the growth period, the plants and pests were managed using on local customs.

Crop evapotranspiration ($ET_c$) under varying irrigation regimes was calculated as follows:

$$ET_c = K_c \times ET_0 \tag{1}$$

where $K_c$ is the crop coefficient. Based on the reference FAO-56 [37], $K_c$ was set at 0.70, 1.00, and 0.95 for the early, middle and late growth stages, respectively. $ET_0$ was calculated by a modified greenhouse Penman-Monteith formula [38,39]:

$$ET_0(P-M) = \frac{0.408\Delta(R_n - G) + \gamma \frac{1713(e_a - e_d)}{T + 273}}{\Delta + 1.64\gamma} \tag{2}$$

where $ET_0$ is the referenced crop evapotranspiration (mm d$^{-1}$), $R_n$ is the surface net radiation (MJ m$^{-2}$ d$^{-1}$), $G$ is the soil heat flux (MJ m$^{-2}$ d$^{-1}$), $e_a$ is the saturated vapor pressure (kPa), $e_d$ is the actual vapor pressure (kPa), $\Delta$ is the slope of the saturated vapor pressure curve (kPa °C$^{-1}$), $\gamma$ is the dry wet constant (kPa °C$^{-1}$), and T is the average temperature at 2 m (°C).

*2.3. Measurements and Calculations*

2.3.1. Morphological Index

During the experiment, five mini Chinese cabbage plants were randomly selected from each plot 45, 60, 75, and 90 days after seeding, and the aboveground plant fresh weight was determined. The plant material was dried at 105 °C for 30 min followed by drying at 75 °C until constant weight. The samples were then cooled in a dryer and weighed using a precision electronic scale. The plant dry weight of each plot was expressed as the average of three plants, and the total aboveground biomass (t ha$^{-1}$) was calculated by multiplying the dry weight by the planting density. The per leaf area (cm$^2$) equals the length multiplied by the width, and the blade shape factor (measured by the grid method) was calculated as 0.69, 0.76, and 0.78 for the early, middle and late growth stages, respectively.

2.3.2. Quality Index

Three plants with similar developmental characteristics were chosen from each plot at harvest time. The vitamin C content was obtained by spectrometry using the molybdenum blue colorimetric method [28]. The nitrate content was obtained using a UV–Vis spectrophotometer (Evolution300, Thermo Fisher Scientific, Waltham, United States).

2.3.3. Water Use Efficiency

The WUE (kg m$^{-3}$) was calculated as follows:

$$WUE = Y/(ET \times 10) \tag{3}$$

where $Y$ is the mini Chinese cabbage yield (t ha$^{-1}$), and $ET$ is the crop water consumption (mm).

$$ET = P_r + I + U - R - D - \Delta W \tag{4}$$

where $P_r$ is the available precipitation (mm); $U$ is the groundwater recharge (mm); $I$ is the amount of irrigation (mm); $R$ is the runoff (mm); $D$ is the deep seepage (mm); and $\Delta W$ is the change in soil moisture from the beginning to the end of the trial (mm). Under the actual conditions during the experiments, the contributions of available precipitation, groundwater recharge, runoff and deep seepage were negligible.

The equation used to calculate $ET$ was therefore:

$$ET = I - \Delta W \tag{5}$$

### 2.3.4. Total N Measurement

After the initial analysis, the aboveground biomass was divided into leaf and leaf stalk. The root samples were gathered at the end of the harvest time. Each separate component of the oven-dried plant material (root, leaf and leaf stalk) was milled to a fine powder. The total N content in each organ was analyzed using a Dumas-type elemental analyzer system (model Rapid N, Elementar, Analysensysteme GmbH, Hanau, Germany). The plants' total N uptake was the sum of the N content uptake in the roots, leaves and leaf stems.

### 2.3.5. Soil NO$_3$-N Content

Soil samples were collected before sowing and after harvesting. Nine sampling points were taken in the test area by using the "Z" type method before planting. The first point was taken from the bottom of the dripper, and the other four points were selected at 10, 20, 40 and 60 cm along the vertical drip tape after harvesting. Five soil layers were sampled at each point: 0–10, 10–20, 20–40, 40–60, and 60–80 cm. The CK soil samples were taken from the middle of the two rows of mini Chinese cabbages by the same point selection method used for the drip fertigation samples. The soil samples were air dried and then passed through a 2 mm sieve. Five grams of soil was weighed and extracted with a 2 mol L$^{-1}$ KCl solution (soil/liquid ratio of 1:10). The content of NO$_3$-N in the soil was determined by a flow analyzer (Auto Analyzer-III, Bran+ Luebbe, Germany). The NO$_3$-N content of each layer was calculated as the average of the five points.

The accumulation of NO$_3$-N (SN, kg ha$^{-1}$) in the soil was calculated as follows:

$$SN = \frac{1}{10}\rho DC \tag{6}$$

where $\rho$ is the soil bulk density (g cm$^{-1}$), $D$ is the soil thickness (cm) and $C$ is the NO$_3$-N content in the soil (mg kg$^{-1}$).

### 2.3.6. Economic Benefits

$$E_b = G - IW - O \tag{7}$$

where $E_b$ is the economic benefits (\$ ha$^{-1}$). $G_R$ is the gross yield (\$ ha$^{-1}$), which is calculated according to the wholesale price in the vegetable market. $I_W$ is the water fee (\$ ha$^{-1}$), which is calculated by multiplying the irrigation amount by the price of water. $O$ includes other inputs (\$ ha$^{-1}$), such as labor costs, irrigation equipment sharing costs, and other costs.

### *2.4. Statistical Analysis*

The value of each indicator was the mean of three replicates per treatment, and SPSS 16 (SPSS, Inc., Chicago, IL, USA) Statistics Software was used to perform analysis of variance. All pairwise

comparisons of the treatment means were performed using the least significant difference (LSD) test, with the significance determined at the 5% level.

## 3. Results

### 3.1. Growth Characteristics

The response of the growth indicators to the irrigation amount and irrigation frequency was consistent in 2014, 2015 and 2016. As the growth period progressed, the aboveground biomass and leaf area per plant gradually increased, and their growth rate was faster in the late growth stage (Figures 1 and 2). Differences began to appear among the treatments approximately 60 days after sowing, and became more apparent thereafter. For the same irrigation amount, the aboveground biomass and leaf area for the $F_2$ irrigation frequency were lower than those for the irrigation frequency but higher than those for the $F_8$ irrigation frequency.

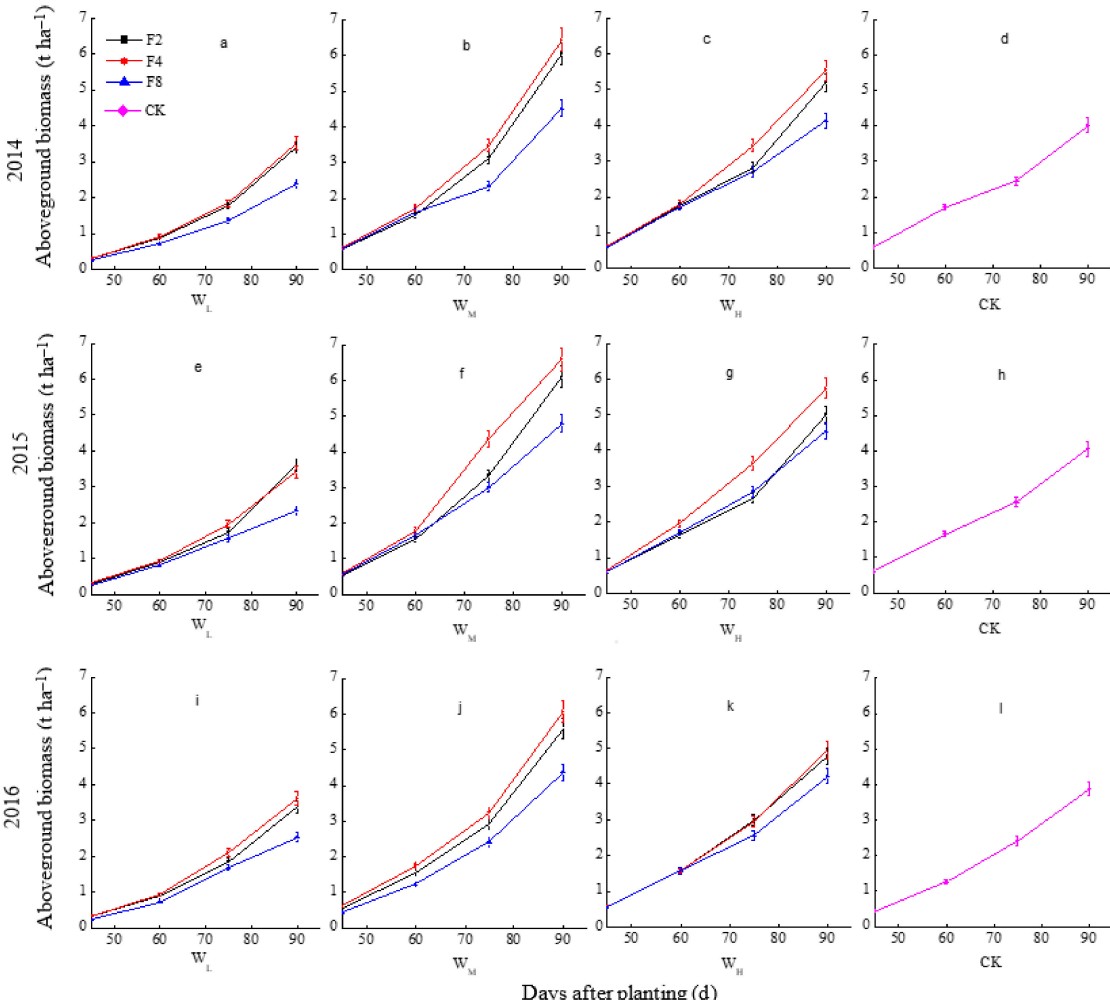

**Figure 1.** Effects of irrigation level and irrigation frequency on aboveground biomass of mini Chinese cabbage in 2014 (**a–d**), 2015 (**e–h**), and 2016 (**i–l**). Irrigation levels: $W_L$: 80% ETc, $W_M$: 120% ETc, $W_H$: 160% ETc. Irrigation frequency: $F_2$: 2 days, $F_4$: 4 days, $F_8$: 8 days, CK, conventional border irrigation with adequate water supply.

For the same irrigation frequency, the aboveground biomass and leaf area for different irrigation amounts increased in the order $W_L < W_H < W_M$. The fastest growth rate was measured for the $W_M F_4$ treatment in all three consecutive years. The maximum aboveground biomass of 6.4, 6.6 and 6.1 g and

the maximum leaf area of 884.8, 921.9 and 892.9 cm$^2$ were obtained approximately 90 days after sowing in 2014, 2015 and 2016, respectively. Compared with conventional border irrigation (CK), the three-year average aboveground biomass and leaf area of $W_M F_4$ were 60% and 54.7% higher, respectively. This treatment can provide enough water and nutrients to the plants and promote their growth.

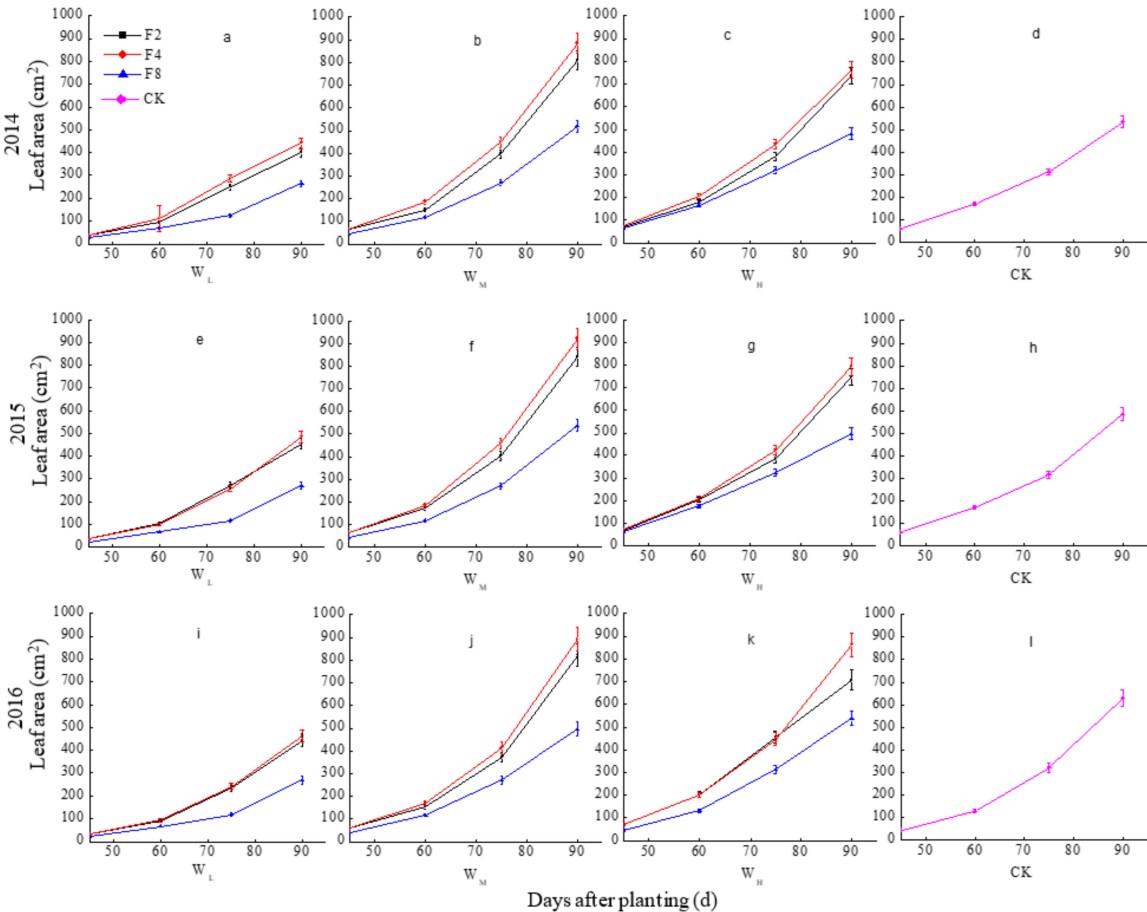

**Figure 2.** Effects of irrigation level and irrigation frequency on the leaf area of mini Chinese cabbage in 2014 (**a–d**), 2015 (**e–h**), and 2016 (**i–l**). Irrigation level, WL: 80% ETc, WM: 120% ETc, WH: 160% ETc. Irrigation frequency, F2: 2 days, F4: 4 days, F8: 8 days. CK, conventional border irrigation with adequate water supply.

### 3.2. Yield and Water Use Efficiency

The effects of irrigation frequency and irrigation level on the yield and WUE in 2014, 2015 and 2016 were extremely significant ($p < 0.01$). The irrigation level had a greater effect on the yield and WUE of the mini Chinese cabbages than did the irrigation frequency. When the irrigation frequency was once every 8 days ($F_8$) and the irrigation amount was low ($W_L$), the yields were 9.8, 8.7 and 8.7 t ha$^{-1}$ in 2014, 2015 and 2016, respectively (Table 2). However, the yield and WUE of the $W_M F_4$ treatment were the highest: the yields were 16.9, 14.5 and 15.9 t ha$^{-1}$, and the WUE values were 29.6, 30.8 and 28.6 kg m$^{-3}$ in 2014, 2015 and 2016, respectively. The lowest WUE was observed in the border irrigation (CK) treatment in 2014 and 2015 and the $W_L F_8$ treatment in 2016. Over the three years, the lowest WUE values were 16.8, 15 and 15.2 kg m$^{-3}$, respectively. Compared with the CK, the three-year average yield and WUE of $W_M F_4$ were increased by 64.5% and 119.2%, respectively. The $W_M F_4$ treatment substantially increased the yield and WUE of the mini Chinese cabbages.

**Table 2.** Effects of irrigation level and irrigation frequency on the yield and WUE of mini Chinese cabbage in 2014, 2015, and 2016. Irrigation level, $W_L$: 80% ETc, $W_M$: 120% ETc, $W_H$: 160% ETc. Irrigation frequency, $F_2$: 2 days, $F_4$: 4 days, $F_8$: 8 days. CK, conventional border irrigation with adequate water supply.

| Treatments | | Yield (t ha$^{-1}$) | | | WUE (kg m$^{-3}$) | | |
|---|---|---|---|---|---|---|---|
| | | **2014** | **2015** | **2016** | **2014** | **2015** | **2016** |
| $W_L$ | $F_2$ | 13.45 ± 0.60 [e] | 13.89 ± 0.62 [f] | 11.85 ± 0.28 [g] | 20.66 ± 0.90 [c] | 21.32 ± 0.93 [c] | 18.5 ± 0.81 [e] |
| | $F_4$ | 12.9 ± 0.58 [e] | 13.55 ± 0.61 [f] | 12.63 ± 0.30 [f] | 20.4 ± 0.25 [c] | 21.43 ± 0.26 [c] | 20.31 ± 0.25 [c] |
| | $F_8$ | 9.79 ± 0.44 [f] | 8.73 ± 0.39 [g] | 8.69 ± 0.20 [h] | 16.83 ± 0.39 [d] | 15.01 ± 0.35 [e] | 15.19 ± 0.36 [f] |
| $W_M$ | $F_2$ | 24.05 ± 1.08 [b] | 26.14 ± 1.17 [b] | 23.34 ± 0.55 [b] | 26.15 ± 1.02 [b] | 28.42 ± 1.11 [b] | 25.81 ± 1.01 [b] |
| | $F_4$ | 27.3 ± 1.23 [a] | 28.34 ± 1.27 [a] | 25.88 ± 0.61 [a] | 29.62 ± 0.58 [a] | 30.76 ± 0.60 [a] | 28.56±0.56 [a] |
| | $F_8$ | 17.79 ± 0.80 [d] | 18.08 ± 0.81 [d] | 15.73 ± 0.37 [e] | 20.55 ± 1.09 [c] | 20.89 ± 1.11 [c] | 18.49 ± 0.98 [e] |
| $W_H$ | $F_2$ | 21.69 ± 0.97 [c] | 22.9 ± 1.03 [c] | 22.5 ± 0.53 [c] | 18.08 ± 0.43 [d] | 19.1 ± 0.46 [d] | 19.08 ± 0.46 [d,e] |
| | $F_4$ | 24.27 ± 1.09 [b] | 23.94 ± 1.08 [c] | 22.95 ± 0.54 [bc] | 21.02 ± 0.63 [c] | 20.73 ± 0.62 [c] | 20.21 ± 0.61 [c,d] |
| | $F_8$ | 18.66 ± 0.84 [d] | 15.96 ± 0.72 [e] | 17.23 ± 0.40 [d] | 16.91 ± 0.65 [d] | 14.47 ± 0.55 [e] | 15.88 ± 0.61 [f] |
| CK | | 16.49 ± 0.36 [d] | 17.07 ± 0.83 [d] | 16.06 ± 0.45 [d] | 13.51 ± 0.81 [e] | 13.93 [d] ± 0.45 [e] | 13.15 ± 0.55 [g] |
| Significance level (F value) | | | | | | | |
| Irrigation amount | | 408.26 ** | 436.97 ** | 1630.63 ** | 246.43 ** | 368.16 ** | 246.62 ** |
| Irrigation Frequency | | 112.42 ** | 194.16 ** | 573.26 ** | 140.38 ** | 271.52 ** | 222.25 ** |
| Amount × Frequency | | 11.16 ** | 6.93 ** | 42.71 ** | 17.81 ** | 7.13 ** | 17.55 ** |

Different letters indicate significant differences among the N fertilizer rates ($p < 0.05$). "*\*" means very significant difference and "\*" means significant difference.

## 3.3. Quality

The irrigation frequency and irrigation amount had significant effects on the nitrate content in 2014, 2015 and 2016 ($p < 0.05$). However, the irrigation frequency and irrigation amount had stronger significant effects on the vitamin C content in the three years ($p < 0.01$). The interaction of irrigation frequency and irrigation amount had an extremely significant effect on the nitrate and vitamin C content (Table 3). Under the same irrigation frequency, the vitamin C content of the mini Chinese cabbages in three irrigation amounts increased in the order $F_8 < F_2 < F_4$, and the cabbages had a significantly higher vitamin C content under $W_M$ than under $W_L$ and $W_H$ ($p < 0.05$).

**Table 3.** Effects of irrigation level and irrigation frequency on the quality of mini Chinese cabbage in 2014, 2015, and 2016. Irrigation level, $W_L$: 80% ETc, $W_M$: 120% ETc, $W_H$: 160% ETc. Irrigation frequency, $F_2$: 2 days, $F_4$: 4 days, $F_8$: 8 days. CK, conventional border irrigation with adequate water supply.

| Treatments | | Nitrate (mg kg$^{-1}$) | | | Vitamin C (mg (100 g)$^{-1}$) | | |
|---|---|---|---|---|---|---|---|
| | | **2014** | **2015** | **2016** | **2014** | **2015** | **2016** |
| $W_L$ | $F_2$ | 278.25 ± 8.43 [c] | 270.93 ± 6.32 [c] | 273.92 ± 3.60 [c] | 42.92 ± 2.50 [f,g] | 42.25 ± 0.75 [f] | 42.75 ± 0.56 [d] |
| | $F_4$ | 306.13 ± 9.05 [b] | 301.56 ± 7.89 [b] | 298.53 ± 10.53 [b] | 44.00 ± 2.64 [d,e,f] | 43.25 ± 0.53 [e] | 43.10 ± 0.35 [d] |
| | $F_8$ | 351.14 ± 7.94 [a] | 346.82 ± 3.93 [a] | 351.17 ± 10.67 [a] | 41.36 ± 2.86 [g] | 40.60 ± 0.45 [g] | 40.05 ± 1.01 [e] |
| $W_M$ | $F_2$ | 251.26 ± 7.45 [e] | 247.55 ± 6.58 [f] | 273.26 ± 8.65 [c] | 48.82 ± 3.35 [b] | 47.60 ± 0.26 [b] | 48.01 ± 0.61 [b] |
| | $F_4$ | 270.10 ± 5.81 [c,d] | 265.83 ± 6.24 [d,e] | 276.05 ± 6.51 [c] | 50.16 ± 3.16 [a] | 48.90 ± 0.50 [a] | 49.25 ± 0.30 [a] |
| | $F_8$ | 290.17 ± 6.39 [c] | 285.66 ± 4.96 [c] | 298.53 ± 9.95 [b] | 44.35 ± 3.31 [d,e] | 43.15 ± 0.58 [e,f] | 43.05 ± 0.73 [d] |
| $W_H$ | $F_2$ | 230.64 ± 7.44 [g] | 225.63 ±7.14 [g] | 213.06 ± 7.65 [e] | 46.77 ± 3.12 [b,c,d] | 45.91 ± 0.31 [c,d] | 45.10 ± 0.79 [c] |
| | $F_4$ | 247.82 ± 7.67 [e,f] | 241.82 ± 6.20 [f] | 235.82 ± 8.36 [d] | 47.57 ± 3.15 [b,c] | 46.75 ± 0.39 [b,c] | 47.25 ± 0.23 [b] |
| | $F_8$ | 261.19 ± 5.75 [d] | 258.61 ± 4.12 [e] | 279.15 ± 13.78 [c] | 44.78 ± 3.55 [d,e] | 43.71 ± 0.92 [e] | 45.20 ± 0.38 [c] |
| CK | | 267.52 ± 6.70 [c,d] | 267.60 ± 1.67 [d,e] | 280.54 ± 3.83 [c] | 46.51 ± 3.02 [b,c,d] | 45.55 ± 0.09 [d] | 44.75 ± 1.34 [c] |
| Significance level (F value) | | | | | | | |
| Irrigation amount | | 5.36 * | 11.22 * | 4.07 * | 102.35 ** | 169.45 ** | 97.50 ** |
| Irrigation Frequency | | 12.23 * | 5.41 * | 9.74 * | 87.19 ** | 127.22 ** | 75.09 ** |
| Amount × Frequency | | 9.06 ** | 12.81 ** | 7.10 ** | 13.25 ** | 9.13 ** | 11.84 ** |

Different letters indicate significant differences among the N fertilizer rates ($p < 0.05$). "*\*" means very significant difference and "\*" means significant difference.

There was a negative correlation between the nitrate content and the amount of irrigation. When the same irrigation frequency was applied, the nitrate content decreased as the amount of irrigation

increased (Table 3). In addition, the nitrate content of $W_L$ was significantly higher than that of $W_M$ and $W_H$ ($p < 0.05$). When the same irrigation amount was applied, the vitamin C content of the mini Chinese cabbages increased in the order $F_8 < F_2 < F_4$; the three-year average vitamin C content of $F_4$ was higher than those of $F_2$ and $F_8$. The nitrate content increased with decreasing irrigation frequency.

### 3.4. Total Nitrogen Content in Different Plant Organs

The irrigation amount and irrigation frequency had significant effects on the N uptake by the organs of the plant. The total nitrogen content of the leaves was the highest, followed by that of the leaf stalk and root. At the same irrigation amount, the total N content of the plants increased in the order $F_8 < F_2 < F_4$ (Figure 3). At the same irrigation frequency, the total N content of the plants increased in the order $W_L < W_H < W_M$. The trend in the total N content between different treatments in the plants was synchronized with the yield of the mini Chinese cabbages. The total N content of the $W_M F_4$ treatment was the highest in all three years, followed by the $W_M F_2$ treatment. The quality of the leaves and leaf stalks was the main determinant of yield, and the leaves and leaf stalks were the primary N-containing vegetative organs of the plants. This result indicated that the total N content of the plants was directly affected by the biomass. Thus, the total N content of $W_M F_4$ was the highest.

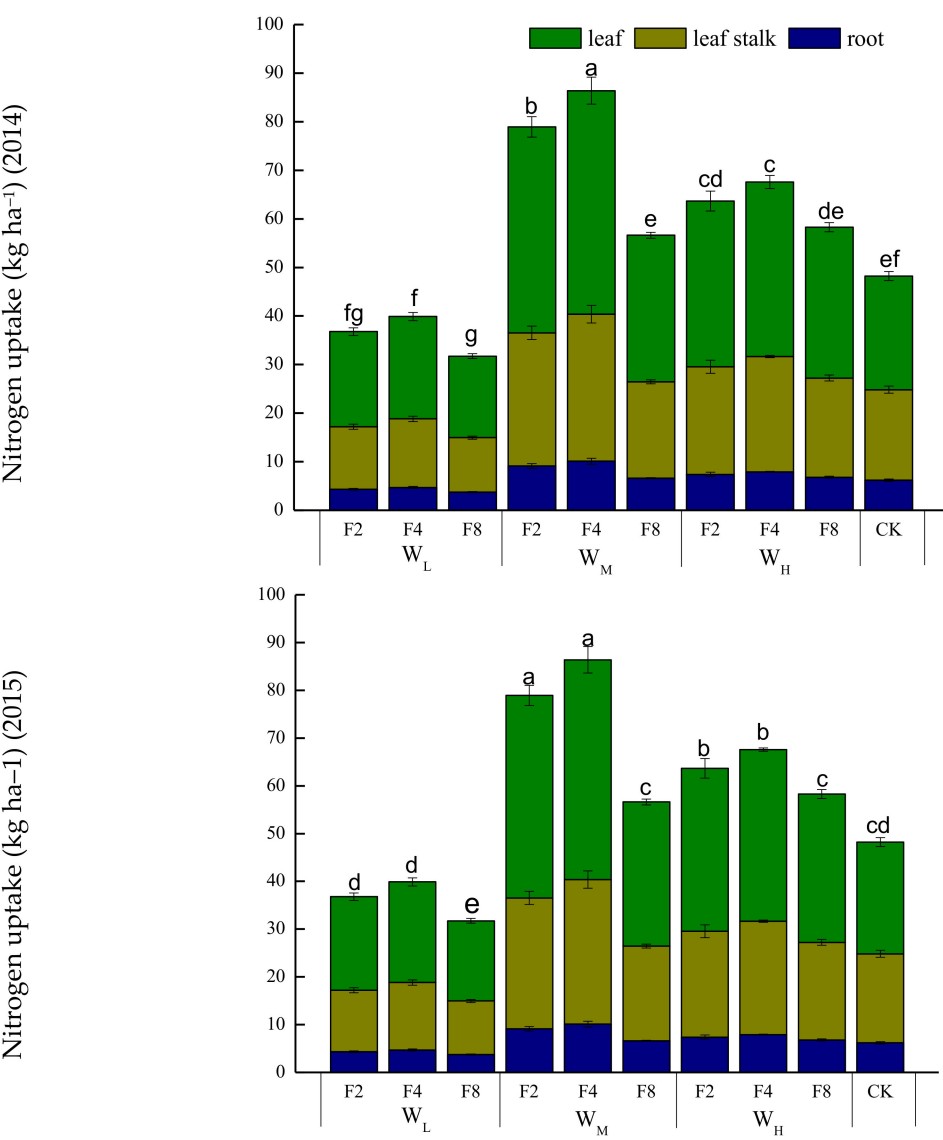

**Figure 3.** *Cont.*

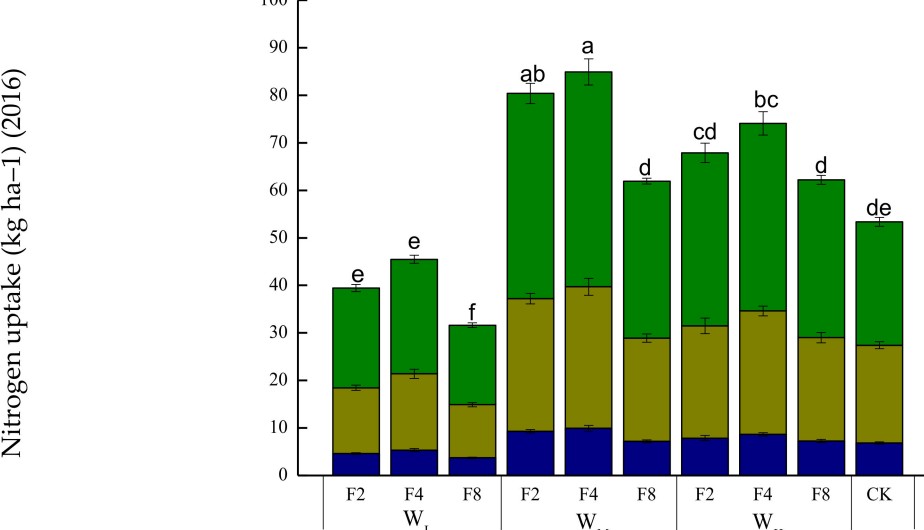

**Figure 3.** Effects of irrigation level and irrigation frequency on total nitrogen uptake in different organs of mini Chinese cabbage in 2014, 2015, and 2016. Irrigation level, $W_L$: 80% ETc, $W_M$: 120% ETc, $W_H$: 160% ETc. Irrigation frequency, $F_2$: 2 days, $F_4$: 4 days, $F_8$: 8 days. CK, conventional border irrigation with adequate water supply. Different letters indicate significant differences among the N fertilizer rates ($p < 0.05$).

### 3.5. The Distribution of Residual Soil $NO_3$-N

Vertical water movement was the main natural factor affecting the distribution of $NO_3$-N in the soil profile. Irrigation increased the water content of the soil profile and was a key human-mediated factor for changing the nitrate content of the soil profile. The distributions of $NO_3$-N in the soil profile before sowing and after harvesting at different irrigation levels and frequencies in 2014, 2015 and 2016 are shown in Figure 4. The vegetables effectively reduced the soil $NO_3$-N content due to absorption by the crop and irrigation during the growing season. After harvesting, the peaks of $NO_3$-N accumulation in the soil profile decreased. The $NO_3$-N content of the 0–40 cm soil profile was significantly lower after harvest than before sowing, and there was no decrease in the $NO_3$-N content in the 40–80 cm soil profile. When the irrigation amount was the same, the peak of residual $NO_3$-N accumulation in the soil profile after harvest decreased with decreasing irrigation frequency. With increasing irrigation amount at the same irrigation frequency, the accumulation peak of $NO_3$-N in the soil profile had a tendency to migrate to the soil layer below the crop root zone after harvest.

The nitrogen balance components of different treatments The nitrogen balance components of different treatments at harvest time in each of the three years are shown in Table 4. Compared to conventional border irrigation (CK), the three-year average soil $NO_3$-N residues of the $W_L$, $W_M$ and $W_H$ increased by 20.96–34.19%, 7.13–13.67%, and 11.84–17.11%, respectively. The N uptake of the plants under the $W_L$ treatment decreased by 15.86% to 38.71%. However, the N uptake in the $W_M$ and $W_H$ treatments increased by 15.89–67.96% and 16.54–39.43%, respectively. In addition, compared to conventional border irrigation (CK), the $NO_3$-N leaching of the $W_L$, $W_M$ and $W_H$ treatments decreased by 5.82–46.21%, 23.19–67.91% and 43.82–72.78%, respectively. The residual $NO_3$-N content in the $W_MF_4$ soil was 1.3% higher than that in the CK soil. However, the N uptake by plants in $W_MF_4$ was 79.16% higher than that of the plants in the CK treatment, and the N loss due to leaching in $W_MF_4$ was 46.33% lower than that in the CK treatment.

**Table 4.** Nitrogen balance components at harvest in 2014, 2015 and 2016 resulting from different irrigation amounts and irrigation frequencies. Irrigation level, $W_L$: 80% ETc, $W_M$: 120% ETc, $W_H$: 160% ETc. Irrigation frequency, $F_2$: 2 days, $F_4$: 4 days, $F_8$: 8 days. CK, conventional border irrigation with adequate water supply.

| Treatments | | Residual N (kg ha$^{-1}$) | Plant N Uptake (kg ha$^{-1}$) | Leached N (kg ha$^{-1}$) | Residual N (kg ha$^{-1}$) | Plant N Uptake (kg ha$^{-1}$) | Leached N (kg ha$^{-1}$) | Residual N (kg ha$^{-1}$) | Plant N Uptake (kg ha$^{-1}$) | Leached N (kg ha$^{-1}$) |
|---|---|---|---|---|---|---|---|---|---|---|
| | | | **2014** | | | **2015** | | | **2016** | |
| $W_L$ | $F_2$ | 107.00 | 36.78 | 26.79 | 117.67 | 34.45 | 19.05 | 127.24 | 39.43 | 20.57 |
| | $F_4$ | 100.62 | 39.89 | 32.72 | 114.56 | 39.13 | 27.30 | 114.05 | 45.49 | 26.97 |
| | $F_8$ | 124.70 | 31.74 | 38.12 | 118.01 | 27.24 | 42.82 | 122.34 | 31.62 | 35.33 |
| $W_M$ | $F_2$ | 104.21 | 78.93 | 13.03 | 95.04 | 71.12 | 15.95 | 100.23 | 80.38 | 10.60 |
| | $F_4$ | 92.07 | 86.39 | 24.26 | 96.99 | 76.71 | 21.93 | 102.30 | 84.92 | 20.06 |
| | $F_8$ | 102.55 | 56.62 | 36.02 | 94.51 | 52.90 | 33.70 | 111.62 | 61.93 | 25.02 |
| $W_H$ | $F_2$ | 101.55 | 63.66 | 17.38 | 115.01 | 59.79 | 6.18 | 98.73 | 67.85 | 10.03 |
| | $F_4$ | 112.66 | 67.59 | 16.11 | 97.54 | 64.55 | 16.53 | 94.37 | 74.07 | 18.23 |
| | $F_8$ | 114.87 | 58.28 | 22.58 | 101.62 | 51.99 | 23.55 | 102.25 | 62.19 | 23.26 |
| CK | | 90.89 | 48.22 | 40.88 | 94.51 | 46.34 | 29.71 | 87.08 | 53.35 | 41.78 |

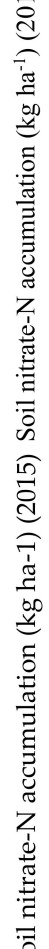

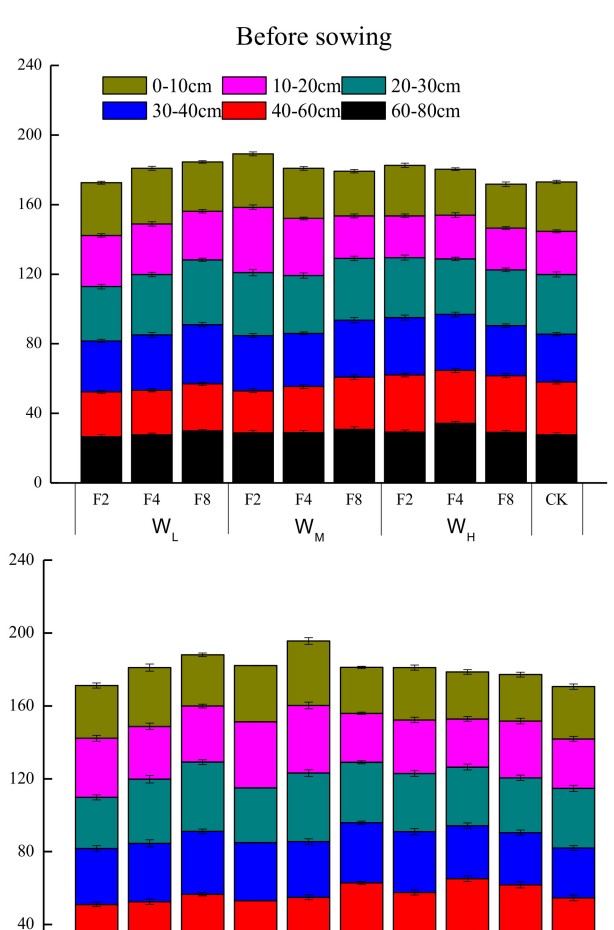

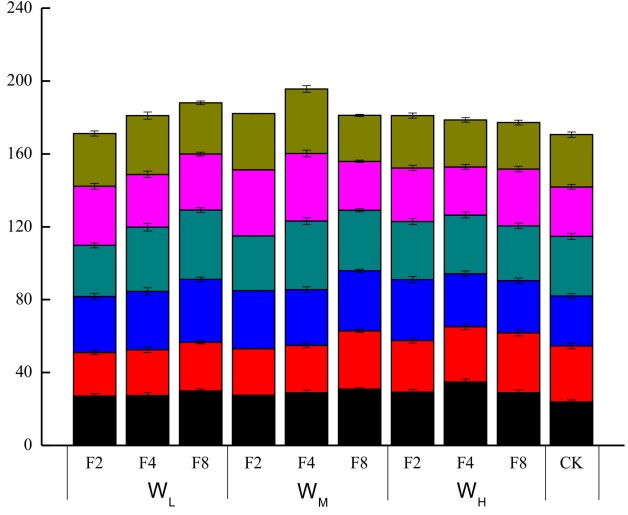

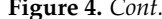

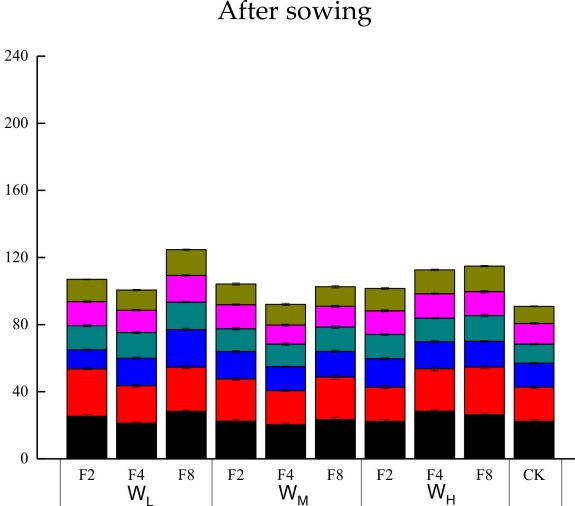

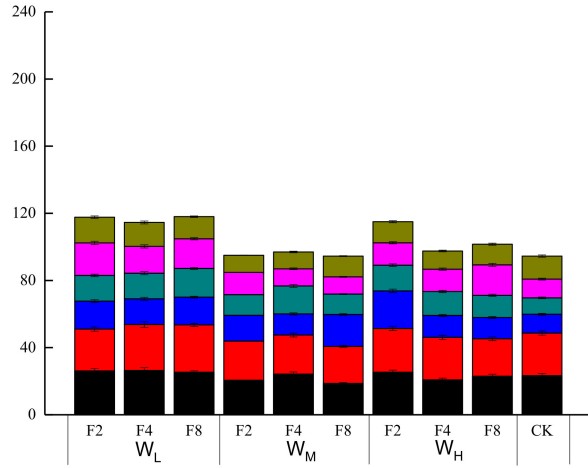

**Figure 4.** *Cont.*

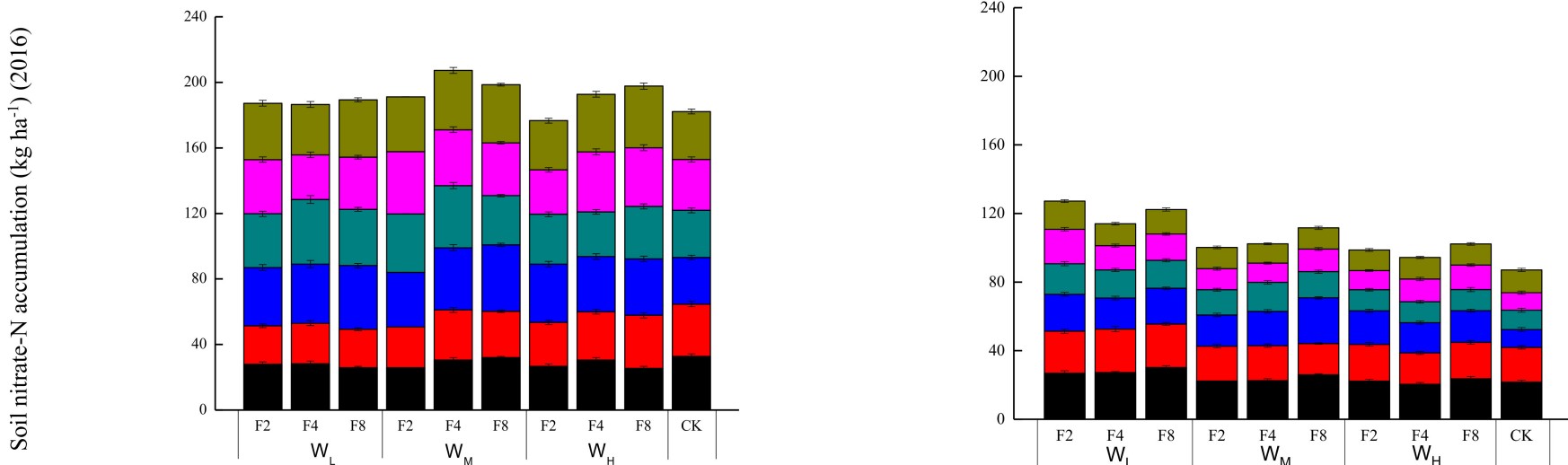

**Figure 4.** Effects of different treatments on the NO$_3$-N concentration in the soil profile in 2014, 2015 and 2016. Irrigation level, W$_L$: 80% ETc, W$_M$: 120% ETc, W$_H$: 160% ETc. Irrigation frequency, F$_2$: 2 days, F$_4$: 4 days, F$_8$: 8 days. CK, conventional border irrigation with adequate water supply.

### 3.6. Economic Benefit Analysis

In addition to water-saving benefits and soil environmental benefits, economic benefits are another important factor in the production of catch crops. The gross gains were 1417–3950 \$ ha$^{-1}$, 1263–4101 \$ ha$^{-1}$ and 1258–3745 \$ ha$^{-1}$ in 2014, 2015 and 2016, respectively (Table 5). Compared to the CK, the gross profits of the low water level ($W_L$) decreased by 18.4%, 18.7% and 26.2%, respectively, and the gross profits of the medium water ($W_M$) and high water ($W_H$) levels increased by different degrees. The minimum economic benefits were 161 \$ ha$^{-1}$, 24 \$ ha$^{-1}$ and 45 \$ ha$^{-1}$, and the maximum economic benefits were 2262 \$ ha$^{-1}$, 2382 \$ ha$^{-1}$ and 2117 \$ ha$^{-1}$ in the three years, respectively. The maximum benefit was observed in the $W_M F_4$ treatment, followed by the $W_M F_2$ treatment. When the low water level was utilized, the gross profits decreased sharply, and the net income was small.

**Table 5.** Effects of different amounts of water and fertilizers on economic benefits.

| Treatments | | Water Fee (\$ ha$^{-1}$) | | | Other Inputs (\$ ha$^{-1}$) | | | Gross Profit (\$ ha$^{-1}$) | | | Economic Benefits (\$ ha$^{-1}$) | | |
|---|---|---|---|---|---|---|---|---|---|---|---|---|---|
| | | 2014 | 2015 | 2016 | 2014 | 2015 | 2016 | 2014 | 2015 | 2016 | 2014 | 2015 | 2016 |
| $W_L$ | $F_2$ | 103 | 109 | 84 | 1233 | 1242 | 1198 | 1947 | 2009 | 1715 | 611 | 658 | 433 |
| | $F_4$ | 103 | 109 | 84 | 1221 | 1235 | 1215 | 1866 | 1960 | 1827 | 543 | 617 | 529 |
| | $F_8$ | 103 | 109 | 84 | 1153 | 1130 | 1129 | 1417 | 1264 | 1258 | 161 | 24 | 45 |
| $W_M$ | $F_2$ | 155 | 164 | 126 | 1463 | 1508 | 1447 | 3480 | 3782 | 3378 | 1863 | 2110 | 1805 |
| | $F_4$ | 155 | 164 | 126 | 1533 | 1556 | 1502 | 3950 | 4101 | 3745 | 2262 | 2382 | 2117 |
| | $F_8$ | 155 | 164 | 126 | 1327 | 1333 | 1282 | 2573 | 2616 | 2277 | 1092 | 1119 | 869 |
| $W_H$ | $F_2$ | 206 | 218 | 168 | 1411 | 1438 | 1429 | 3138 | 3314 | 3256 | 1521 | 1658 | 1659 |
| | $F_4$ | 206 | 218 | 168 | 1467 | 1460 | 1439 | 3513 | 3464 | 3320 | 1839 | 1786 | 1714 |
| | $F_8$ | 206 | 218 | 168 | 1345 | 1287 | 1315 | 2699 | 2310 | 2493 | 1148 | 805 | 1011 |
| CK | | 362 | 420 | 325 | 1299 | 1311 | 1289 | 2386 | 2470 | 2324 | 725 | 739 | 710 |

Irrigation level, $W_L$: 80% ETc, $W_M$: 120% ETc, $W_H$: 160% ETc. Irrigation frequency, $F_2$: 2 days, $F_4$: 4 days, $F_8$: 8 days. CK, conventional border irrigation with adequate water supply.

Compared to that of the CK, the water fee of the medium irrigation level ($W_M$) was reduced by 208 \$ ha$^{-1}$, 256 \$ ha$^{-1}$ and 199 \$ ha$^{-1}$ in 2014, 2015 and 2016, respectively (Table 5). The proportion of the water fee in the total investment was low. Appropriately increasing the amount of irrigation water could significantly increase the net income. However, the increase in the net income of mini Chinese cabbages was not significant when water was applied more frequently. This pattern is why farmers were misled into thinking that "the greater the irrigation amount and frequency, the greater the income". When the medium ($W_M$) and high irrigation ($W_H$) levels were applied, the net income increased in the order $F_8 < F_2 < F_4$. Thus, high-frequency irrigation was not an effective method of increasing economic benefits.

## 4. Discussion

### 4.1. Growth Characteristics, Yield, Quality, Water Use Efficiency and Economic Benefits

Scientific water management is the key to obtaining high yields and saving irrigation water throughout the stages of crop growth [40]. An appropriate irrigation regime can significantly improve crop growth, resulting in increased economic yields. Soil moisture and nutrient conditions were directly affected by the irrigation regime, and a positive correlation between crop growth and soil moisture in a certain range was observed [41]. When irrigation water can be used in a timely manner to offset the water consumption of crops, the crops can grow and develop rapidly. However, too much water can lead to extended oversaturation of soil moisture in the root zone, resulting in the inability of the root system to respire normally, thus inhibiting the growth of the plants, whereas insufficient water cannot meet the normal water requirements of the crops, resulting in slow growth [42]. A lower

irrigation frequency and the use of more water per irrigation can increase the risk of soil $NO_3$-N leaching under certain irrigation conditions [43]. Infrequent irrigation and insufficient expansion of the wet ellipsoids are not conducive to nutrient uptake by root systems. Appropriate irrigation times can ensure that the wet ellipsoids extend to the root growth area when each irrigation is applied without producing excess runoff, resulting in an optimal crop root zone soil environment and improvement in crop root growth by enlarging the contact area of the roots and soil, stimulating the movement of water and nutrients, and enhancing the absorption of water and N nutrition by the crops [44]. Under this condition, the crop leaf extension rate would be increased, and assimilation would be greatly enhanced due to the wide leaf area of a single plant [45,46].

The water supply during growth is one of the main factors influencing crop quality. The analytical results of this study indicated that when the irrigation frequency was constant, increased irrigation water led to a decrease in the nitrate content in plants to a certain extent and an increase in the vitamin C content. When the irrigation amount was constant, a decrease in irrigation frequency led first to an increase in nitrate and vitamin C content and then to a decrease. The results showed that the quality of the mini Chinese cabbages was the highest at the appropriate irrigation amount and appropriate irrigation frequency ($W_MF_4$ treatment). Reasonable upper and lower limits of irrigation frequency benefit crop growth and improve crop quality [47,48].

The analytical results of this study showed that the irrigation amount and frequency both significantly influenced the yield and WUE of mini Chinese cabbages, and the irrigation amount had a greater effect on the yield and WUE of mini Chinese cabbages than did the frequency. The 3-year test results showed that the yield and WUE of the $F_4$ treatment were higher than those of the $F_2$ and $F_8$ treatments at the same irrigation level. The $W_M$ treatment yield and WUE were higher than those in the $W_L$ and $W_H$ treatments at the same irrigation frequency. The maximum values of both the yield and WUE were obtained in the $W_MF_4$ treatment under the interaction of irrigation amount and frequency.

Achieving economic benefits is another of the main objectives of fallow crop production. The results show that the application of drip irrigation technology in a certain range can significantly improve the green economic benefits by increasing the amount and frequency of irrigation. However, excessive irrigation frequency or irrigation amount may reduce economic benefits to some extent. In contrast, the economic benefit of the $W_MF_4$ treatment was 210.8% than the control on average, that of the $W_MF_2$ treatment was 165.6% higher, and that of the $W_HF_4$ treatment was 145.5% higher on average.

## 4.2. Irrigation Levels and Irrigation Frequencies Adjust the Soil $NO_3$-N Distribution

The total loss from nitrification and denitrifying N in the greenhouse soil was negligible [49,50]. Some of the $NO_3$-N retained by the former crops in the greenhouse soil was absorbed by the plants, some leached with the irrigation water, and some remained in the soil. The residual $NO_3$-N and plant uptake under each treatment were higher than those under the CK treatment under drip irrigation conditions, while the leaching loss was less than that under the CK treatment. When the irrigation amount was the same, the irrigation frequency decreased, and the irrigation amount per time increased, which led to the increase in residual $NO_3$-N in the 40–80 cm soil layer. Thus, the total amount of residual $NO_3$-N in the soil increased. The amount of leaching also increased, while the N absorbed by the plants decreased. At the same irrigation frequency, the accumulation peaks of residual $NO_3$-N in the soil varied in response to the increase in the amount of irrigation, but the difference in the total amount was small. The amount of N uptake by the plants increased in the order $W_L < W_H < W_M$. However, the amount of N leaching decreased as the amount of irrigation increased. This effect was mainly due to the higher amounts of residual $NO_3$-N under the low irrigation level ($W_L$), whereas the amount of plant absorption was higher under the medium irrigation level ($W_M$). The amount of leached N was affected not only by the soil $NO_3$-N before sowing and after harvesting, but also by the absorption by the plants.

The irrigation amount and frequency of irrigation water application directly affect the transport of soil $NO_3$-N and residual soil $NO_3$-N [51–53]. Nitrogen uptake and utilization by crop roots also

affected nitrate transport and accumulation in the soil [54]. When irrigated with excessive water, the soil will be saturated for a long time, and the aeration conditions will become worse because of the formation of an anaerobic environment, which hinders the occurrence of reverse digestion [55]. Simultaneously, a large amount of gravity-driven water infiltration intensifies the migration of soil nitrate, which exists in the form of nitrate ions in high-humidity soil and continuously dissolves the solid nitrate in the soil during the downward migration process, resulting in an increase in its concentration [56–58]. With a shortage of irrigation water and a lack of soil moisture, $NO_3$-N primarily forms solids and is not mobile in the soil. Under conditions with the same amount of irrigation, the quantity of each single irrigation quantity increases as the irrigation frequency decreases, resulting in an improvement in the vertical infiltration of soil moisture and an increase in the soil $NO_3$-N vertical movement over those produced by dripped water. The profile of nitrate accumulation in the soil is thus decreased and extended. The 0 to 40 cm soil layer is the active region of the primary root system of vegetables under drip irrigation, which maximizes the absorption and utilization of the soil $NO_3$-N. In the 40-80 cm soil layer, $NO_3$-N is transported with water and not absorbed by the root system of the crops, and the residual $NO_3$-N content after harvest is large. Under the same irrigation frequency, the total accumulation of soil $NO_3$-N does not peak in response to an increase in irrigation water, and the amount of N leaching decreases with increasing irrigation water. These processes occur mainly because the crop can take up more water and nutrients when an appropriate irrigation regime ($W_M F_4$ treatment) is applied. The amount of N leaching is affected by the amount of residual $NO_3$-N both before and after the sowing of mini Chinese cabbages and even by the crop uptake. It can be seen that increasing the irrigation frequency and reducing the amount of water in each irrigation does not provide adequate water, which is not conducive to crop growth. In contrast, reducing the irrigation frequency and increasing the single irrigation amount can easily lead to the accumulation of $NO_3$-N in the underlying soil, increasing the risk of $NO_3$-N leaching and the potential risk of groundwater pollution. According to the analysis conducted in this study, an appropriate irrigation amount and irrigation frequency ($W_M F_4$ treatment) are conducive to promoting the absorption of water and nutrients by crop roots, leaving minimal residual soil nitrate and $NO_3$-N. Greenhouse is an important facility for the vegetable production around the world due to its high-efficient use of solar radiation resources [59–61]. In summary, moderate irrigation amounts and frequency ($W_M F_4$ treatment) are conducive to promoting the normal growth and development of mini Chinese cabbages, effectively improving the yield, WUE and quality, as well as significantly reducing the soil residual $NO_3$-N content, which is conducive to the sustainable use of greenhouse soil.

## 5. Conclusions

In the cultivation and management of mini Chinese cabbages, the amount and frequency of irrigation determine the soil moisture and soil moisture depth around plants and directly affect the yield, quality and soil $NO_3$-N migration associated with mini Chinese cabbages. The $W_M F_4$ treatment showed no significant change in the soil residual $NO_3$-N (only a 1.3% increase) compared to the alternate treatment and no significant change in the accumulation peak over the three-year period of this study. Compared to the CK treatment, the $W_M F_4$ treatment increased plant N uptake by 68% (33.4 kg ha$^{-1}$), decreased N leaching loss by 40% (15.4 kg ha$^{-1}$), increased aboveground biomass by 60% (2.7 t ha$^{-1}$), increased leaf area by 54.7% (484.7 cm$^2$), increased yield by 64% (10.6 t ha$^{-1}$), and increased WUE by 119.1% (16.1 kg m$^3$), averaged over the three years. These results indicate that the $W_M F_4$ treatment can offer a reasonable soil moisture environment for mini Chinese cabbages and achieve the goals of high yield, high quality and high efficiency, while reducing the risk of soil nitrate leaching. Furthermore, this treatment did not significantly increase the amount of $NO_3$-N accumulation in the soil.

Under these experimental conditions, the $W_M F_4$ treatment (an irrigation amount of 120% ETc and a frequency of every 4 days) can be recommended as a more reasonable irrigation system under fallow greenhouse management measures in the Guanzhong area. This study provides an irrigation model

with high quality, high yield and high-efficiency N regulation for the planting of fall-winter crops in the Guanzhong area and provides a new theoretical basis for the sustainable utilization, optimization and management of greenhouse soil in this area.

**Author Contributions:** Y.X., H.Z., X.W. and F.Z. conceived and designed the experiments; S.Q., Y.W., S.Y., H.W., L.W., and J.F. performed the experiments; Y.X., H.Z. and X.W. analyzed the data; F.Z. contributed reagents/materials/analysis tools; X.W. wrote the paper.

**Funding:** This research was funded by the National Key Research and Development Program of China (No. 2017YFC0403303), the National Natural Science Foundation of China (51579211, 51809224), Shaanxi Innovation Capacity Support Project (2018KJXX-080, 2018JQ5218) and Key Project of Education Department of Shaanxi Province (18JS117).

**Conflicts of Interest:** The authors declare no conflict of interest.

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
