# Peer review of "Effect of Irrigation Level and Irrigation Frequency on the Growth of Mini Chinese Cabbage and Residual Soil Nitrate Nitrogen"

_sustainability, doi:10.3390/su11010111_

Round 1

Reviewer 1 Report

Overall the manuscript provides an interesting scope by trying to advice farmers with the best water strategy in terms of ecological benefit and not wasting water by intense irrigation.

My two major criticism are:

- You don’t really mentioned the fertilization strategy you were using. How much NO3 did you used in this experiment, how much is recovered in the plant or how many from this is really lost? I don’t see anything about this in the method, result or discussion section. Your whole manuscript starts in the introduction about how bad N fertilizer can be, but after that you never mention it again. I think you have to restructure the introduction with a better focus on the water supply or give a better overview about the fertilizer use in your experiment. Also in Table 4 suddenly some amount of fertilizer are mentioned again, without any idea where this is coming from.

- Your conclusion is write, but the numbers you are presenting are totally misleading or just wrong. You make it look like the highest number are the average, but you cannot do this.

Also it would be good if you already talk about % values to give an average and look if this is also statistically significant. Especially in the part about NO3-N.

Other minor comments:

I would use NO3-N in the manuscript and not always write nitrate-N, especially after you wrote it after L196.

In the method section the part about the Total N content measurement is missing. Also the description which parts of the plant were used for this measurement is missing.

You have to explain the calculations for the economic benefit better, especially since you referring in the result section to the Chinese currency, without mentioning before how you calculated this. 

L24 Don’t use abbreviations in the abstract if you don’t explain them (ETc)

L26-28 Shorten this sentence: “The results showed that the 26 weight of the leaves and leaf stalks was the primary determinant of yield, and the leaves and 27 leaf stalks were also the primary N-containing vegetative organs of the plants. “ to  “The results showed that the weight of the leaves and leaf stalks was the primary determinant of yield and also the primary N-containing vegetative organs of the plants. “

L34 add the between in WmF4

L40 delete the

L51 add N after kg

L51-53 I think this sentence is Grammarly not correct

L53-54 Also in this sentence if you talk about the past use were instead of are. Also if you use vegetables as plural don’t refer to it as it, use they

L54-55 Also this sentence should be restructured to: “Particularly in the Guanzhong region, the agricultural facilities in this region have developed rapidly in the past two decades.”

L61-64 You should also mention the effect on the emission of N2O

L69-72 Split this sentence in two

L69 Which catch crops and what are catch crops?

L83 Why is it increasing especially during Spring Festival?

L93 Use the plural from Research

L95 delete , and so on.

L113 Change of the to in

L117-121 would be nicer to have that in a Table form.

L127- 149 Did you use any fertilizer?

 L128 You should add were in front of obtained.

L135 write also the first 10 in words. Also you can add the second sentence by just writing: “in a total of ten treatments, with three replicates in a randomized complete block factorial design.”

L142 What does The stubble mean?

L160 delete by incubation

L167 No italic font

L168-171 Which machines were used?

L204 There is something wrong with the formatting

L208 No italic font

L210 Which version of SPSS was used, maybe a little bit more description of the statistical analysis.

L223-224 and 230-231 What do you mean with first increased and then decreased? I cannot see any decrease in any of your graphs?

Fig 1. If you show the Irrigation level from low – high, use also the same order in the text or vice versa (check this also through the whole manuscript). And the explanation for CK (also for all following Fig. and Tables) is missing

L231-233 Please restructure the sentence and begin with: The fastest growth rate was measured… or something like that

Fig. 2 Is not about aboveground biomass.

L 268 I would use: stronger significant effects instead of extremely. Also following I wouldn’t use the term extremely significant, especially if you never describe this before in the method section.

L 280 Was it really significant different in all your treatments? Because for many treatments it looks like that they are not so different, especially with the overlapping STDEV. At least from F2 to F4 the differences seem not so strong.

L280 delete However

Table 2. Please make this Table in a proper way. Either make the text smaller or delete one decimal number, so everything fits in the columns.

L288 Use different plant organs

L290-291 Don’t use three times total N content

L299 Change indicated to indicate

Fig. 3 the axis name and the legend is partly cut off.

L312 Change to soil nitrate-N residues

Table 3. Same as Table 2

L330-332 I don’t see any NO3 in Table 3, I think you referring to the Residual, Plant and Leached N, you should than use also NO3-N. I checked also some of your % calculations and received most of the times different numbers, than the ones you are mentioning. E.g. for WL residual N you have 100.62, 107.00 and 124.70 kg /ha, while you have 90.89 at the CK treatment

100.62 are an increase of just 10.07%

While 124.70 is an increase of 37.2%

So please check all your calculations again. Or provide prove that your calculations are correct.

L332 First mention the Table than show it in the manuscript

L339-342 This is more a part for the conclusion and not the result section

Fig. 4 The axis at year 2015 is cut off again. Different font in the left and the right part

L345 Put the Fig. 4 before this text

L353 What is RMB?

Table 4 I cannot really read anything in this table, because the formatting is wrong.

L398 From your data I don’t really see this decrease of aboveground biomass, it increased over the complete incubation.

L419-422 This sentence is too long and that makes it difficult to understand it.

L427-428 You cannot say something about the loss if you did not measured the N2O+N2 fluxes.

L467-468 I don’t understand the term single irrigation water soil

L480-  Check all the numbers again, because also here you are talking about 79.16% increase in plant N uptake, but in the results you showed only 15.89%-67.96%. This is either wrong or totally misleading or just cheating. If you state the highest numbers only, than write that in the conclusion.

L494-496 redundant and wrong

Author Response

Mr. Wang Xiukang

Yan'an University, Yan'an, Shaanxi 716000, China

wangxiukang@126.com

December 17, 2018

Dear Professor Sean Tate and anonymous reviewers:

The authors greatly appreciate the valuable, detailed and professional comments on this paper.

As the corresponding author, I received an e-mail informing me that our manuscript (Ref: 388938, “Characteristic Response of the Growth of Greenhouse-Grown Mini Chinese Cabbage and Residual Soil Nitrate Nitrogen to Different Irrigation Levels and Irrigation Frequencies”) has been reviewed, with the reviewer comments included. We were very glad to receive your e-mail. Thank you again for your letter and for giving us the opportunity to revise our manuscript. The authors are deeply grateful to all of you for your support and your careful review of our manuscript. We have studied the comments carefully and have made corrections, which we hope meet with your approval. The main corrections in the paper and our responses to the comments are given below.

Comments and Suggestions for Authors

Overall the manuscript provides an interesting scope by trying to advice farmers with the best water strategy in terms of ecological benefit and not wasting water by intense irrigation.

Response: Thank you very much for your comments. We greatly appreciate your putting so much energy into reviewing this paper and providing so many valuable suggestions.

My two major criticism are:

- You don’t really mentioned the fertilization strategy you were using. How much NO3 did you used in this experiment, how much is recovered in the plant or how many from this is really lost? I don’t see anything about this in the method, result or discussion section. Your whole manuscript starts in the introduction about how bad N fertilizer can be, but after that you never mention it again. I think you have to restructure the introduction with a better focus on the water supply or give a better overview about the fertilizer use in your experiment. Also in Table 4 suddenly some amount of fertilizer are mentioned again, without any idea where this is coming from.

Response: Thank you very much for your valuable comments. We have added the text “The crop preceding this experiment was sweet pepper in 2014 and 2015 and tomato in 2016. The same fertilizer level (N400–P2O5225–K2O150 kg ha−1), chosen on the basis of the amount of fertilizer commonly used locally, was applied to the sweet pepper and tomato. No fertilizer was applied throughout the growth period of mini Chinese cabbage” in this revision, please check it. Thank you very much.

- Your conclusion is write, but the numbers you are presenting are totally misleading or just wrong. You make it look like the highest number are the average, but you cannot do this. Also it would be good if you already talk about % values to give an average and look if this is also statistically significant. Especially in the part about NO3-N.

Response: Thank you very much for your valuable comments. We have added the values and retained the % values as follows: “Compared to the CK treatment, the WMF4 treatment increased plant N uptake by 68% (33.4 kg ha−1), decreased N leaching loss by 40% (15.4 kg ha−1), increased aboveground biomass by 60% (2.7 t ha−1), increased leaf area by 54.7% (484.7 cm2), increased yield by 64% (10.6 t ha−1), and increased WUE by 119.1% (16.1 kg m3), averaged over the three years.”. Thank you again for your valuable advice. 

Other minor comments:

I would use NO3-N in the manuscript and not always write nitrate-N, especially after you wrote it after L196.

Response: Thank you very much for your support and valuable comments. We have revised “nitrate-N” to “NO3-Nthroughout the manuscript.

In the method section the part about the Total N content measurement is missing. Also the description which parts of the plant were used for this measurement is missing.

Response: Thank you very much for your support and valuable comments. We have added the following text “2.3.4. Total N measurement. After the initial analysis, the aboveground biomass was divided into leaf and leaf stalk. The root samples were gathered at the end of the harvest time. Each separate component of the oven-dried plant material (root, leaf and leaf stalk) was milled to a fine powder. The total N content in each organ was analyzed using a Dumas-type elemental analyzer system (model Rapid N, Elementar, Analysensysteme GmbH, Hanau, Germany). The plants’ total N uptake was the sum of the N content uptake in the roots, leaves and leaf stems” in this revision, please check it. Thank you again for your valuable advice.

You have to explain the calculations for the economic benefit better, especially since you referring in the result section to the Chinese currency, without mentioning before how you calculated this.

Response: Thank you very much for your support and valuable comments. We have added the following text: “2.3.6. Economic benefits

Eb=GR-IW-O                           (7)

where Eb is the economic benefits ($ ha−1). GR is the gross yield ($ ha−1), which is calculated according to the wholesale price in the vegetable market. IW is the water fee ($ ha−1), which is calculated by multiplying the irrigation amount by the price of water. O includes other inputs ($ ha−1), such as labor costs, irrigation equipment sharing costs, and other costs.In addition, we have changed the Chinese currency to United States dollars in this revision. Thank you again for your valuable advice.

L24 Don’t use abbreviations in the abstract if you don’t explain them (ETc)

Response: Thank you very much for your support and valuable comments. We have added “WH: 160% crop evapotranspiration (ETc)” to the Abstract in this revision.

L26-28 Shorten this sentence: “The results showed that the 26 weight of the leaves and leaf stalks was the primary determinant of yield, and the leaves and 27 leaf stalks were also the primary N-containing vegetative organs of the plants.” to “The results showed that the weight of the leaves and leaf stalks was the primary determinant of yield and the primary N-containing vegetative organs of the plants.”

Response: We are particularly grateful to you for helping us revise this sentence. We have revised the sentence “The results showed that the weight of the leaves and leaf stalks was the primary determinant of yield, and the leaves and leaf stalks were also the primary N-containing vegetative organs of the plants” to “The results showed that the weight of the leaves and leaf stalks was the primary determinant of yield and that these are the primary N-containing vegetative organs of the plants..” Thank you again for your valuable advice.

L34 add the between in WmF4

Response: We are particularly grateful to you for helping us revise this sentence. The sentence has been rewritten more extensively in this revision.

L40 delete the

Response: We are particularly grateful to you for helping us revise this sentence. We have deleted “the” from line 40 in this revision. Thank you again for your valuable advice.

L51 add N after kg

Response: We are particularly grateful to you for helping us revise this sentence. We have added “N” after kg in line 51 in this revision.

L51-53 I think this sentence is Grammarly not correct

Response: Thank you very much for your support and valuable comments. We have revised this sentence to “The amount of N fertilizer applied each year is almost three times the amount absorbed by vegetables.”. Thank you again for your valuable advice.

L53-54 Also in this sentence if you talk about the past use were instead of are. Also if you use vegetables as plural don’t refer to it as it, use they

Response: We are particularly grateful to you for helping us revise this sentence. We have revised this sentence to “In recent years, vegetables have been the main crops grown in greenhouses in China because they provide consumers with many health benefits and have high economic value.” Thank you again for your valuable advice.

L54-55 Also this sentence should be restructured to: “Particularly in the Guanzhong region, the agricultural facilities in this region have developed rapidly in the past two decades.”

Response: We are particularly grateful to you for helping us revise this sentence. We have revised this sentence to “Particularly in the Guanzhong region, agricultural facilities have developed rapidly in the past two decades.”

L61-64 You should also mention the effect on the emission of N2O

Response: Thank you very much for your support and valuable comments. We have added “including simultaneous nitrification and denitrification by ammonia volatilization and N2O and N2 emissions” in this revision.

L69-72 Split this sentence in two

Response: Thank you very much for your support and valuable comments. We have split this sentence to “The roots of catch crops such as ryegrass and fodder radish can effectively absorb the remaining NO3-N in the soil. Moreover, after harvest, the soil NO3-N accumulation of 0-1.0 m deep soil planted with leeks and red beets was lower than that of soil without catch crops.”

L69 Which catch crops and what are catch crops?

Response: Thank you very much for your support and valuable comments. We have identified the catch crops as follows: “The roots of catch crops such as ryegrass and fodder radish can effectively absorb the remaining NO3-N in the soil.”

L83 Why is it increasing especially during Spring Festival?

Response: Thank you very much for your support and valuable comments. In China, the economic status of most rural people is not high. Particularly in winter, people mainly eat vegetables that were dried in summer. In the Spring Festival, many relatives and friends visit each other and will find good things to entertain guests. It is better to serve vegetables in winter because they are more expensive, and it is the custom in China to show hospitality by offering something expensive. Since other readers may be unfamiliar with this regional culture, we have modified this sentence to “As vegetables are healthful foods, the demand for vegetables has increased significantly in recent years.” Thank you again for your valuable advice.

L93 Use the plural from Research

Response: Thank you very much for your support and valuable comments. We have revised “Research has shown” to “Studies have shown” in this sentence. We particularly appreciate your help in improving the quality of our manuscript.

L95 delete , and so on.

Response: Thank you very much for your support and valuable comments. We have deleted “, and so on” from the text.

L113 Change of the to in

Response: Thank you very much for your support and valuable comments. We have changed “of the” to “in” in this revision.

L117-121 would be nicer to have that in a Table form.

Response: Thank you very much for your support and valuable comments. We have presented the information in Table 1 in this revision.

Table 1 Major soil physicochemical characteristics (BD, bulk density; OM, organic matter; TN, total nitrogen; TP, total phosphorus; TK, total potassium; AP, available phosphorus; AK, available potassium) of the experimental site measured before the experiment (n=5).

Year

Soil depth (cm)

BD

(g cm−3)

OM

(g kg−1)

pH

TN

(g kg−1)

TP

(g kg−1)

TK

(g kg−1)

AP

(mg kg−1)

AK

(mg kg−1)

2014

0-20

1.42

14.33

8.2

0.62

0.59

12.8

34.2

101.2

20-40

1.38

15.44

8.4

0.76

0.52

16.8

21.6

97.6

40-60

1.49

11.66

8.3

0.87

0.49

16.7

26.7

107.1

60-80

1.43

13.59

8.3

0.71

0.53

15.5

13.5

91.5

2015

0-20

1.40

16.63

8.0

0.71

0.68

14.8

18.7

114.6

20-40

1.47

15.88

8.3

0.65

0.53

15.6

15.8

103.3

40-60

1.44

14.52

8.3

0.86

0.52

17.1

14.6

88.6

60-80

1.37

13.49

8.1

0.79

0.42

16.8

12.6

94.5

2016

0-20

1.38

14.54

8.2

0.87

0.49

11.6

22.1

109.2

20-40

1.44

15.87

8.2

0.79

0.56

14.7

16.7

115.8

40-60

1.49

13.15

8.0

0.66

0.54

16.2

17.4

110

60-80

1.40

12.98

8.1

0.75

0.46

14.4

10.9

94.4

L127- 149 Did you use any fertilizer?

Response: Thank you very much for your support and valuable comments. We have added “No fertilizer was applied throughout the growth period of mini Chinese cabbage.” in this revision.

L128 You should add were in front of obtained.

Response: Thank you very much for your support and valuable comments. We have added “were” before “obtained” in this revision.

L135 write also the first 10 in words. Also you can add the second sentence by just writing: “in a total of ten treatments, with three replicates in a randomized complete block factorial design.”

Response: We are particularly grateful to you for helping us revise it. We have added the text (CK, WHF2, WHF4, WHF8, WMF2, WMF4, WMF8, WLF2, WLF4 and WLF8). Three replicates of each treatment were performed in a randomized complete block factorial design.”. Thank you again for your valuable advice.

L142 What does The stubble mean?

Response: Thank you very much for your support and valuable comments. Stubble means the previous crop. We have revised “the stubble” to “the previous crop” for clarity. Thank you again for your valuable advice.

L160 delete by incubation

Response: Thank you very much for your support and valuable comments. We have deleted “by incubation” from this revision. Thank you again for your valuable advice.

L167 No italic font

Response: Thank you very much for your support and valuable comments. We have revised the use of italics.

L168-171 Which machines were used?

Response: Thank you very much for your support and valuable comments. We have added “UV–Vis spectrophotometer (Evolution300, Thermo Fisher Scientific, Waltham, United States)” in this revision. Thank you again for your valuable advice.

L204 There is something wrong with the formatting

Response: Thank you very much for your support and valuable comments. We have corrected the formatting.

L208 No italic font

Response: Thank you very much for your support and valuable comments. We have revised the use of italics.

L210 Which version of SPSS was used, maybe a little bit more description of the statistical analysis.

Response: Thank you very much for your support and valuable comments. We have added the version of SPSS and revised to the text as follows: “The value of each indicator was the mean of three replicates per treatment, and SPSS 16 (SPSS, Inc., United States) Statistics Software was used to perform analysis of variance.”

L223-224 and 230-231 What do you mean with first increased and then decreased? I cannot see any decrease in any of your graphs?

Response: Thank you very much for your support and valuable comments. We have revised it the text as follows: “For the same irrigation amount, the aboveground biomass and leaf area for the F2 irrigation frequency were lower than those for the irrigation frequency but higher than those for the F8 irrigation frequency” and “For the same irrigation frequency, the aboveground biomass and leaf area for different irrigation amounts increased in the order WL < WH < WM.” Thank you again for your valuable advice.

Fig 1. If you show the Irrigation level from low – high, use also the same order in the text or vice versa (check this also through the whole manuscript). And the explanation for CK (also for all following Fig. and Tables) is missing.

Response: Thank you very much for your support and valuable comments. We have revised the text to “Irrigation levels: WL: 80% ETc, WM: 120% ETc, WH: 160% ETc. Irrigation frequency: F2: 2 days, F4: 4 days, F8: 8 days, CK, conventional border irrigation with adequate water supply throughout the manuscript. We greatly appreciate your help in improving the quality of our manuscript.

L231-233 Please restructure the sentence and begin with: The fastest growth rate was measured… or something like that

Response: Thank you very much for your support and valuable comments. We have revised the text to “The fastest growth rate was measured for the WMF4 treatment in all three consecutive years.”

Fig. 2 Is not about aboveground biomass.

Response: Thank you very much for your support and valuable comments. We sincerely apologize for our carelessness. We have revised “aboveground biomass” to “leaf area.”

L 268 I would use: stronger significant effects instead of extremely. Also following I wouldn’t use the term extremely significant, especially if you never describe this before in the method section.

Response: Thank you very much for your support and valuable comments. We have revised it to “stronger significant effects” in this revision, please check it. We particularly appreciate your help in improving the quality of our manuscript.

L 280 Was it really significant different in all your treatments? Because for many treatments it looks like that they are not so different, especially with the overlapping STDEV. At least from F2 to F4 the differences seem not so strong.

Response: Thank you very much for your support and valuable comments. We have revised this sentence to “When the same irrigation amount was applied, the vitamin C content of the mini Chinese cabbages increased in the order F8 < F2 < F4; the three-year average vitamin C content of F4 was higher than those of F2 and F8.” We greatly appreciate your help in improving the quality of our manuscript.

L280 delete However

Response: Thank you very much for your support and valuable comments. We have deleted “however” in this revision.

Table 2. Please make this Table in a proper way. Either make the text smaller or delete one decimal number, so everything fits in the columns.

Response: Thank you very much for your support and valuable comments. We have made the text smaller in this revision.

L288 Use different plant organs

Response: Thank you very much for your support and valuable comments. We have revised “different organs” to “different plant organs.”.

L290-291 Don’t use three times total N content

Response: Thank you very much for your support and valuable comments. We have revised the text to “The total nitrogen content of the leaves was the highest, followed by that of the leaf stalk and root.”

L299 Change indicated to indicate

Response: Thank you very much for your support and valuable comments. We have changed “indicated” to “indicate” in this revision.

Fig. 3 the axis name and the legend are partly cut off.

Response: Thank you very much for your support and valuable comments. We have corrected this error.

L312 Change to soil nitrate-N residues

Response: Thank you very much for your support and valuable comments. We have revised “soil nitrate nitrogen residues” to “residual soil NO3-N.”

Table 3. Same as Table 2

Response: Thank you very much for your support and valuable comments. We have made the recommended changes.

L330-332 I don’t see any NO3 in Table 3, I think you referring to the Residual, Plant and Leached N, you should than use also NO3-N. I checked also some of your % calculations and received most of the times different numbers, than the ones you are mentioning. E.g. for WL residual N you have 100.62, 107.00 and 124.70 kg /ha, while you have 90.89 at the CK treatment. 100.62 are an increase of just 10.07%. While 124.70 is an increase of 37.2%. So please check all your calculations again. Or provide prove that your calculations are correct.

Response: Thank you very much for your support and valuable comments. We used the three-year average value of residual soil NO3-N in the previous revision. The calculation process is as follows:

2014

WL:

F2: Growth rate of residual N = (107-90.89)/90.89

                        =17.72%

F4: Growth rate of residual N = (100.62-90.89)/90.89

                        =10.71%

F8: Growth rate of residual N = (124.7-90.89)/90.89

                        =37.2%

WM:

F2: Growth rate of residual N = (104.21-90.89)/90.89

                        =14.66%

F4: Growth rate of residual N = (92.07-90.89)/90.89

                        =1.3%

F8: Growth rate of residual N = (102.55-90.89)/90.89

                        =12.83%

WH:

F2: Growth rate of residual N = (101.55-90.89)/90.89

                        =11.73%

F4: Growth rate of residual N = (112.66-90.89)/90.89

                        =23.95%

F8: Growth rate of residual N = (114.87-90.89)/90.89

                        =26.38%

2015

WL:

F2: Growth rate of residual N = (117.67-94.51)/94.51

                        =24.51%

F4: Growth rate of residual N = (114.56-94.51)/94.51

                        =21.21%

F8: Growth rate of residual N = (118.01-94.51)/94.51

                        =24.87%

WM:

F2: Growth rate of residual N = (95.04-94.51)/94.51

                        =0.56%

F4: Growth rate of residual N = (96.99-94.51)/94.51

                        =2.62%

F8: Growth rate of residual N = (94.51-94.51)/94.51

                        =0

WH:

F2: Growth rate of residual N = (115.01-94.51)/94.51

                        =21.69%

F4: Growth rate of residual N = (97.54-94.51)/94.51

                        =3.21%

F8: Growth rate of residual N = (101.62-94.51)/94.51

                        =7.52%

2016

WL:

F2: Growth rate of residual N = (127.24-87.08)/87.08

                        =46.12%

F4: Growth rate of residual N = (114.05-87.08)/87.08

                        =20.97%

F8: Growth rate of residual N = (122.34-87.08)/87.08

                        =40.49%

WM:

F2: Growth rate of residual N = (100.23-87.08)/87.08

                        =15.1%

F4: Growth rate of residual N = (102.3-87.08)/87.08

                        =17.48%

F8: Growth rate of residual N = (111.62-87.08)/87.08

                        =28.18%

WH:

F2: Growth rate of residual N = (98.73-87.08)/87.08

                        =13.38%

F4: Growth rate of residual N = (94.37-87.08)/87.08

                        =8.37%

F8: Growth rate of residual N = (102.25-87.08)/87.08

                        =17.42%

Therefore, we can calculate the increased residual N under WL as follows:

F2: increase = (F2: Growth rate of residual N in 2014 + 2015 +2016) / 3

= (17.72+24.51+46.12) / 3

= 29.45%

F4: increase = (F4: Growth rate of residual N in 2014 + 2015 +2016) / 3

= (10.71+21.21+30.97) / 3

= 20.96%

F8: increase = (F8: Growth rate of residual N in 2014 + 2015 +2016) / 3

= (37.20+24.87 +40.49) / 3

= 34.19%

WM increased the residual N as follows:

F2: increased = (F2: Growth rate of residual N in 2014 + 2015 +2016) / 3

= (14.66+0.56+15.10) / 3

= 10.11%

F4: increased = (F4: Growth rate of residual N in 2014 + 2015 +2016) / 3

= (1.30+2.62+17.48) / 3

= 7.13%

F8: increased = (F8: Growth rate of residual N in 2014 + 2015 +2016) / 3

= (12.83+0.00+28.18) / 3

= 13.67%

WH increased the residual N as follows:

F2: increase = (F2: Growth rate of residual N in 2014 + 2015 +2016) / 3

= (11.73+21.69 +13.38) / 3

= 15.60%

F4: increase = (F4: Growth rate of residual N in 2014 + 2015 +2016) / 3

= (23.95+3.21+8.37) / 3

= 11.84%

F8: increase = (F8: Growth rate of residual N in 2014 + 2015 +2016) / 3

= (26.38+7.52+7.42) / 3

= 17.11%

From the above, the three-year average soil NO3-N residues under the WL, WM and WH treatments increased by 20.96%-34.19%, 7.13%-13.67%, and 11.84%-17.11%, respectively.

We thank you again for your valuable advice. We used the average of three years to reduce the error of one year's results.

L332 First mention the Table than show it in the manuscript

Response: Thank you very much for your support and valuable comments. We have added “The nitrogen balance components of different treatments at harvest time in each of the three years are shown in Table 3.”

L339-342 This is more a part for the conclusion and not the result section

Response: Thank you very much for your support and valuable comments. We have removed this sentence from the Results section and incorporated it into the Conclusion. We greatly appreciate your help in improving the quality of our manuscript.

Fig. 4 The axis at year 2015 is cut off again. Different font in the left and the right part

Response: Thank you very much for your support and valuable comments. We have corrected this error.

L345 Put the Fig. 4 before this text

Response: Thank you very much for your support and valuable comments. We have made the recommended correction.

L353 What is RMB?

Response: Thank you very much for your support and valuable comments. We have changed the Chinese currency to United States dollars in this revision. Thank you again for your valuable advice.

Table 4 I cannot really read anything in this table, because the formatting is wrong.

Response: Thank you very much for your support and valuable comments. We have revised the table. Thank you again for your valuable advice.

L398 From your data I don’t really see this decrease of aboveground biomass, it increased over the complete incubation.

Response: Thank you very much for your support and valuable comments. We want to compare the results of different treatments at the same sample collection time. The aboveground biomass tends to increase first and then decrease with increasing irrigation frequency. However, this result has little to do with the topic of this manuscript. To prevent ambiguity, we have deleted this sentence.

L419-422 This sentence is too long and that makes it difficult to understand it.

Response: Thank you very much for your support and valuable comments. We have revised to the sentence as follows: “The results show that the application of drip irrigation technology in a certain range can significantly improve the green economic benefits by increasing the amount and frequency of irrigation. However, excessive irrigation frequency or irrigation amount may reduce economic benefits to some extent.”

L427-428 You cannot say something about the loss if you did not measured the N2O+N2 fluxes.

Response: Thank you very much for your support and valuable comments. We have revised it to “The total loss from nitrification and denitrifying N in the greenhouse soil was negligible.” We greatly appreciate your help in improving the quality of our manuscript.

L467-468 I don’t understand the term single irrigation water soil

Response: Thank you very much for your support and valuable comments. We have revised this sentence to “It can be seen that increasing the irrigation frequency and reducing the amount of water in each irrigation does not provide adequate water, which is not conducive to crop growth” in this revision, please check it. We greatly appreciate your help in improving the quality of our manuscript.

L480- Check all the numbers again, because also here you are talking about 79.16% increase in plant N uptake, but in the results, you showed only 15.89%-67.96%. This is either wrong or totally misleading or just cheating. If you state the highest numbers only, than write that in the conclusion.

Response: Thank you very much for your support and valuable comments. We have checked all the numbers in this revision. We have added the values and retained the % values in this revision.

L494-496 redundant and wrong

Response: Thank you very much for your support and valuable comments. We have deleted this sentence from the revised manuscript. We are very grateful to you for spending so much time on this manuscript to help us improve the quality of the paper. All the authors sincerely thank you for your professional and meticulous help.

We have tried our best to improve the manuscript and have made several changes. However, these changes have not influenced the content or framework of the paper. We greatly appreciate the work of the Editor and reviewers, and we hope that our revision will meet with approval.

In addition, we send the paper to AJE for language modification. Please check the proof of language modification.

Once again, thank you very much for your comments and suggestions.

Kind regards,

Dr. Xiukang Wang

Phone: +86-911-2332030

Fax: +86-911-2332030

Email: wangxiukang@126.com

Reviewer 2 Report

line 69. indicate the level in china and compare with the level in other countries

lines 102 to 106. Improve the text. I think that the idea is too much theoretical and not an objective

line 117 check the soil texture (only 1% of sand?)

lines 127 to 133  Why use a different irrigation system as control. I think you should take a 100% ETc as control. It is difficult to compare two different systems.

line 151. Equation (2) is the typical Penman-Monteith equation. Why the citation is Wang et al , 2006, instead of Penman, XX cited by Wang et al, XX ?

On the other hand citation system is wrong, from cite (1) to (33) appears only the number and the cites are ordered by the order of appearing in the text. Suddenly apears cites following the clasic stile (Allen et al, 1998) etc, but these cites are not in the references chapter. In the chapter discussion and sucesives, the cites sistem return to the initial system. (33) to (52). Please Use only one system and reference properly the cites.

lines 176 to 185 equations (3), (4) and (5) lack of units

line 204. put a carriage return.

Results. It should be better to explain what is the best tretreatment inted of say that "fist increased and then decreased" (line 271 for example)

Improve table 3. sometimes a word is split or displaced

line 353 What is RMB ?. I think it is not defined previously

line 407 "...to the former research... At this point this text is confuse. Please indicate clearly what is this topic or cite the sameone.

Author Response

Mr. Wang Xiukang

Yan'an University, Yan'an, Shaanxi 716000, China

wangxiukang@126.com

December 17, 2018

Dear Professor Sean Tate and anonymous reviewers:

The authors greatly appreciate the valuable, detailed and professional comments on this paper.

As the corresponding author, I received an e-mail informing me that our manuscript (Ref: 388938, “Characteristic Response of the Growth of Greenhouse-Grown Mini Chinese Cabbage and Residual Soil Nitrate Nitrogen to Different Irrigation Levels and Irrigation Frequencies”) has been reviewed, with the reviewer comments included. We were very glad to receive your e-mail. Thank you again for your letter and for giving us the opportunity to revise our manuscript. The authors are deeply grateful to all of you for your support and your careful review of our manuscript. We have studied the comments carefully and have made corrections, which we hope meet with your approval. The main corrections in the paper and our responses to the comments are given below.

Comments and Suggestions for Authors

Line 69. indicate the level in china and compare with the level in other countries

Response: Thank you very much for your support and valuable comments. The World Health Organization (WHO) has set the maximum permissible limit of NO3-N to 8 mg L-1. We have revised the text as follows: “The relative content of NO3-N in the 1-2.0 m soil layer of some test sites is more than 40% [17], and the content of NO3-N in the groundwater of some areas is close to the limit of drinking water sanitation standards (World Health Organization, 8 mg L–1). The content of NO3-N in the groundwater has approached the permissible limit of 6.4 mg L–1 in the 5.25 -5.5 m soil layer in the Guanzhong region [18]. The roots of catch crops such as ryegrass and fodder radish can effectively absorb the remaining NO3-N in the soil [19]. Moreover, after harvest, the soil NO3-N accumulation of 0-1.0 m deep soil planted with leeks and red beets was lower than that of soil without catch crops [19, 20].

lines 102 to 106. Improve the text. I think that the idea is too much theoretical and not an objective

Response: Thank you very much for your support and valuable comments. We have revised to the text as follows: “Answering these questions is essential to recommend a suitable irrigation regime for the use of mini Chinese cabbage in fallow greenhouse management in northwest China. In particular, this information should facilitate the establishment of scientific greenhouse catch crop irrigation systems, help to maintain the sustainable use of soil, and provide a theoretical basis for the sustainable development of agriculture.” We greatly appreciate your help in improving the quality of our manuscript.

line 117 check the soil texture (only 1% of sand?)

Response: Thank you very much for your support and valuable comments. We have revised this sentence. We greatly appreciate your help in improving the quality of our manuscript.

lines 127 to 133 Why use a different irrigation system as control. I think you should take a 100% ETc as control. It is difficult to compare two different systems.

Response: Thank you very much for your support and valuable comments. We will use this suggested experimental design in our future experiments. Thank you very much for your advice.

line 151. Equation (2) is the typical Penman-Monteith equation. Why the citation is Wang et al, 2006, instead of Penman, XX cited by Wang et al, XX ?

Response: Thank you very much for your support and valuable comments. Equation (2) is a modified greenhouse Penman-Monteith formula. We have revised the reference (Penman, H. L., Natural evaporation from open water, bare soil and grass. Proceedings of the Royal Society of London 1948, 193, (1032), 120-145).

On the other hand citation system is wrong, from cite (1) to (33) appears only the number and the cites are ordered by the order of appearing in the text. Suddenly apears cites following the clasic stile (Allen et al, 1998) etc, but these cites are not in the references chapter. In the chapter discussion and sucesives, the cites sistem return to the initial system. (33) to (52). Please Use only one system and reference properly the cites.

Response: Thank you very much for your support and valuable comments. We sincerely apologize for our carelessness. We have revised the reference style throughout the manuscript.

lines 176 to 185 equations (3), (4) and (5) lack of units

Response: Thank you very much for your support and valuable comments. We have added the units in this revision.

line 204. put a carriage return. 

Response: Thank you very much for your support and valuable comments. We have made this correction.

Results. It should be better to explain what is the best tretreatment inted of say that "fist increased and then decreased" (line 271 for example)

Response: Thank you very much for your support and valuable comments. We have revised to the text as follows: “Under the same irrigation frequency, the vitamin C content of the mini Chinese cabbages in three irrigation amounts increased in the order F8 < F2 < F4, and the cabbages had a significantly higher vitamin C content under WM than under WL and WH (P < 0.05).” We also fixed several similar problems throughout the manuscript. We greatly appreciate your help in improving the quality of our manuscript.

Improve table 3. sometimes a word is split or displaced

Response: Thank you very much for your support and valuable comments. We have corrected Table 3 in this revision.

line 353 What is RMB ?. I think it is not defined previously

Response: Thank you very much for your support and valuable comments. We have changed the currency to “$” in this revision.

line 407 "...to the former research... At this point this text is confuse. Please indicate clearly what is this topic or cite the sameone.

Response: Thank you very much for your support and valuable comments. We have revised the text to “Reasonable upper and lower limits of irrigation frequency benefit crop growth and improve crop quality.

We have tried our best to improve the manuscript and have made several changes. However, these changes have not influenced the content or framework of the paper. We greatly appreciate the work of the Editor and reviewers, and we hope that our revision will meet with approval.

In addition, we send the paper to AJE for language modification. Please check the proof of language modification.

Once again, thank you very much for your comments and suggestions.

Kind regards,

Dr. Xiukang Wang

Phone: +86-911-2332030

Fax: +86-911-2332030

Email: wangxiukang@126.com

Round 2

Reviewer 1 Report

Dear Authors,

I think your manuscript definitely improved with all the changes. I would approve it for publication.

Kindest regards

Reviewer 2 Report

the authors have improved their work in many respects. It is an interesting work and I hope it will also be of interest for readers.